# Denoising Normalizing Flow

**Christian Horvat**
Department of Physiology
University of Bern
Bern, Switzerland
`christian.horvat@unibe.ch`

**Jean-Pascal Pfister**
Department of Physiology
University of Bern
Bern, Switzerland
`jeanpascal.pfister@unibe.ch`

## Abstract

Normalizing flows (NF) are expressive as well as tractable density estimation methods whenever the support of the density is diffeomorphic to the entire data-space. However, real-world data sets typically live on (or very close to) low-dimensional manifolds thereby challenging the applicability of standard NF on real-world problems. Here we propose a novel method - called Denoising Normalizing Flow (DNF) - that estimates the density on the low-dimensional manifold while learning the manifold as well. The DNF works in 3 steps. First, it inflates the manifold - making it diffeomorphic to the entire data-space. Secondly, it learns an NF on the inflated manifold and finally it learns a denoising mapping - similarly to denoising autoencoders. The DNF relies on a single cost function and does not require to alternate between a density estimation phase and a manifold learning phase - as it is the case with other recent methods. Furthermore, we show that the DNF can learn meaningful low-dimensional representations from naturalistic images as well as generate high-quality samples.

## 1   Introduction

Given samples from the data-density $p(x)$, key objectives in probabilistic Machine Learning are 1. estimating $p(x)$ *(density estimation)*, 2. generating new data points from $p(x)$ *(sampling)*, and 3. finding low-dimensional representations of the data *(inference)*. The three main methods used to perform these tasks are *Normalizing Flows* (NFs) [34], *Generative Adversarial Networks* (GANs) [17], and *Variational Autoencoders* (VAEs) [26]. Among those methods, only the VAE does check all desired objectives by default. However, it does so by sacrificing the sample quality (compared to GANs), and only learning a lower bound on $p(x)$ (rather than the exact value as NFs do). Finding new ways to meet these key objectives, based on either the known main methods or new ones, is an active and important research area.

How can we infer low-dimensional representations using NFs? The standard NF requires the data-density $p(x)$ to have a support diffeomorphic to the entire data-space $\mathbb{R}^D$. However, the manifold hypothesis conjectures that the data-manifold $\mathcal{M}$ lies close to a $d$-dimensional manifold embedded in $\mathbb{R}^D$, $d < D$. Thus, unfortunately, a standard NF cannot be used to infer the latent space. Recently, Brehmer et al. proposed to overcome this limitation by first projecting the data into a $d$-dimensional space using the first $d-$components of a standard NF, and then using another NF to learn the latent distribution $\pi(u)$ [10]. For their method to work they propose different learning schemes, separating the manifold learning from the density estimation.

In this paper, we propose an easy and new way to learn low-dimensional representations with NFs. Our idea is based on the theoretical work derived in [20] where it was shown that by inflating the data-manifold with Gaussian noise $\varepsilon \sim \mathcal{N}(0, \sigma^2 I_D)$, i.e. $\tilde{x} := x + \varepsilon$, the data-density $p(x)$ can be well approximated by learning the inflated distribution $q_\sigma(\tilde{x})$. More concretely, the main result in [20] states sufficient conditions on the choice of noise $q_\sigma(\tilde{x}|x)$ and type of manifold such that

$q_\sigma(x) = p(x)q_\sigma(x|x)$ holds. Here, by adding a penalty term to the usual KL-divergence used to learn $q_\sigma(\tilde{x})$, we ensure that the first $d$-components of the corresponding flow are noise insensitive and thus encode the manifold. This penalty term is essentially the objective function of a *Denoising autoencoder* (DAE), and thus we call our method *Denoising Normalizing Flow* (DNF).

In summary, our contributions are the following:

1. We propose a new method, the Denoising Normalizing Flow, which combines two previously well-known methods (DAE and NF), and is able to
   - approximate $p(x)$,
   - sample new data $x \sim p(x)$ as a non-linear transformation of $u \sim \mathcal{N}(u; 0, I_d)$,
   - infer low-dimensional latent variables $u \sim p(u|x)$ given $x \sim p(x)$.
2. We demonstrate on naturalistic data that our method learns meaningful latent representations without sacrificing the sample quality.

**Notations:** We adapt the notation used in [10]. To further simplify it and avoid clutter, we denote the Gram matrix of $g$ evaluated at $g^{-1}(x)$ as

$$G_g(x) := J_g(g^{-1}(x))^T J_g(g^{-1}(x)) \tag{1}$$

where $J_g(g^{-1}(x))^T$ is the transpose of the Jacobian of $g : \mathbb{R}^d \to \mathbb{R}^D$, $d \leq D$, evaluated at $g^{-1}(x)$.

## 2 Problem statement

In the following, we are going to show why standard NFs are not suited to infer low-dimensional representations of the given data. We end the section with the research question we are going to study in Section 3.

**General setting:** In generative modeling, it is assumed that the data $x \in \mathcal{M} \subset \mathbb{R}^D$ is generated by a non-linear transformation $g$ of some latent variables $u \in \mathcal{U} \subset \mathbb{R}^d$, i.e. $x = g(u)$ where $u \sim \pi(u)$. Typically, $d < D$ and the latent distribution $\pi(u)$ is assumed to be Gaussian, $\pi(u) = \mathcal{N}(u; 0, I_d)$. Hence, the latent random variable $u$ generates $x$, and the data-density $p(x)$ evaluated at $x \in \mathbb{R}^D$ is given by

$$p(x) = \int_{\mathcal{U}} \pi(u)\delta(x - g(u))du, \tag{2}$$

where $\delta$ denotes the Dirac function, see [3]. If $d = D$ and $g$ is a diffeomorphism, we have for $x \in \mathcal{M}$ that

$$p(x) = |\det G_g(x)|^{-\frac{1}{2}} \pi(g^{-1}(x)). \tag{3}$$

Then, the target density $p(x)$ can be learned, in principle, exactly using an NF [21]. In general, an NF is an embedding mapping $\mathcal{U} \subset \mathbb{R}^d$ to $\mathcal{M} \subset \mathbb{R}^D$, see [32] or [27] for some recent reviews. Denoting this mapping as $g_\theta : \mathcal{U} \to \mathcal{M}$ and its parameters as $\theta$, the induced density on $\mathcal{M}$ is given by

$$p_\theta(x) = |\det G_{g_\theta}(x)|^{-\frac{1}{2}} p_u(g_\theta^{-1}(x)), \tag{4}$$

where $p_u(u)$ is a known reference density (usually set to be standard Gaussian). The parameters $\theta$ are updated such that the KL-divergence between $p(x)$ and $p_\theta(x)$,

$$D_{\mathrm{KL}}(p(x)||p_\theta(x)) = -\mathbb{E}_{x \sim p(x)}[\log p_\theta(x)] + const. \tag{5}$$

is minimized. Thus, to learn $\theta$ efficiently, one needs to evaluate $T_1(x) := \log p_u(g_\theta^{-1}(x))$ and $T_2(x) := \frac{1}{2}\log|\det G_{g_\theta}(x)|$ efficiently.

**Topological constraints:** The evaluation of $T_1(x)$ is efficient since we are free to choose the reference measure $p_u$, and $g_\theta^{-1}(x)$ is the forward pass of a neural network constructed to be bijective [13, 21]. Therefore, the majority of the NF literature focuses on designing clever flow architectures to be able to calculate $T_2(x)$ efficiently without sacrificing the flow's expressiveness (i.e. the size of the space of embeddings able to learn). However, so far these architectures are constructed for $d = D$ since in this case, the Jacobian of $g_\theta$ is a square matrix, and thus $T_2(x)$ becomes

$$\frac{1}{2}\log|\det G_{g_\theta}(x)| = \log|\det J_{g_\theta}(g_\theta^{-1}(x))|. \tag{6}$$

Hence, calculating the Gram determinant of $g_\theta$ efficiently amounts to calculating the determinant of the Jacobian of $g_\theta$ efficiently. Popular choices are to construct $g_\theta$ such that $J_{g_\theta}$ is lower triangular, as in this case, the determinant is simply the product of diagonal elements. Unfortunately, for $d < D$, $J_{g_\theta}$ is not a square matrix, and the full Gram determinant needs to be calculated. This makes NFs unsuitable for finding low-dimensional representations $u$ of high-dimensional data points for large $d$ as the computational complexity to calculate $\det G_{g_\theta}$ is $\mathcal{O}(d^2 D) + \mathcal{O}(d^3)$.

**Research question:** As mentioned in [10], finding ways to design $g_\theta$ such that $T_2(x)$ can be efficiently calculated for $d < D$ is an interesting research question. Here, we address it from a different angle.

*Let $\mathcal{M}$ be a $d-$dimensional manifold embedded in $\mathbb{R}^D$ through $g$. Can we construct a mapping $g_\theta : \mathbb{R}^d \to \mathcal{M}$ to*

1. *generate $x \sim p(x)$ in 2 steps: (a) generate $u \sim \mathcal{N}(u; 0, I_d)$ and (b) set $x = g_\theta(u)$?*

2. *infer $u \in \mathbb{R}^d$ such that $x = g_\theta(u)$?*

3. *approximate $\det G_g(x)$ efficiently?*

## 3   Denoising Normalizing Flow

We answer the research question based on the theoretical work developed in [20]. First, we briefly review this work, and then introduce the DNF.

**Preliminaries:** In Section 2, we discussed why classical NFs are not suited to infer low-dimensional representations of the given data. Also, if one is only interested in the value of $p(x)$, standard NF cannot be used. Intuitively, a standard NF $f_\psi$ with parameters $\psi$ simply squeezes or expands a volume element where the net change is given by its Jacobian determinant. A volume element of a $d-$dimensional manifold is $d-$dimensional and thus has $D-$dimensional Lebesgue measure 0. Thus, we are asking $f_\psi$ to expand a $d-$dimensional volume to a $D-$dimensional one which will lead to a degeneration of $|\det G_{f_\psi}(x)|$ and manifest in numerical instabilities. Therefore, [20] inflated the manifold by adding Gaussian noise to the data points.[1] This inflated manifold is $D-$dimensional and thus a usual flow can be used to learn the corresponding density. Their main result states sufficient conditions on the choice of noise and type of manifold $\mathcal{M}$, such that the learned inflated distribution can be deflated, and $p(x)$ is exactly retrieved.

More precisely, given a random variable $x \sim p(x)$ with probability measure $\mathbb{P}_X$ and taking values in a $d-$dimensional manifold $\mathcal{M}$ embedded in $\mathbb{R}^D$, if we add some noise $\varepsilon$ to it, the resulting new random variable $\tilde{x} = x + \varepsilon$ has the following density:

$$q_\sigma(\tilde{x}) = \int_{\mathcal{M}} q_\sigma(\tilde{x}|x) d\mathbb{P}_X(x). \tag{7}$$

In [20], it was shown that if (a) the noise is only added in the $(D-d)-$dimensional normal space $N_x$ in $x$, (b) the noise magnitude $\sigma$ is sufficiently small, and (c) the manifold $\mathcal{M}$ is sufficiently smooth and disentangled, the resulting inflated distribution evaluated at $\tilde{x} = x$ takes the following product form:

$$q_\sigma(x) = p(x) q_\sigma(x|x), \tag{8}$$

where $q_\sigma(x|x)$ is the normalization constant of the noise density. More concretely, (a) and (b) need to ensure that $x$ is almost surely uniquely determined by $\tilde{x}$ as the orthogonal projection of $\tilde{x}$ on $\mathcal{M}$. For this projection to be well-defined, a sufficient condition is that the manifolds reach number[2] is finite [6]. A manifold where almost every point $\tilde{x} \sim q(\tilde{x})$ in the inflated set has a unique projection on $\mathcal{M}$ was called $Q-$normally reachable in [20], where $Q$ denotes the collection of noise distributions $q_\sigma(\tilde{x}|x)$. Their main Theorem proves that for any $Q-$normally reachable manifold equation (8) holds. Therefore, if $q_\sigma(\tilde{x})$ can be learned exactly using a standard NF, the on-manifold density $p(x)$ can be retrieved exactly.

It was also shown that for the case where $d \ll D$ (as it is generally assumed for high-resolution images), full Gaussian noise is an excellent approximation for a Gaussian in the normal space.

---

[1] Adding noise to circumvent the aforementioned degeneracy problem was also proposed in [24].

[2] Informally, this reach condition ensures that a manifold is learnable through samples.

**Main idea:** We use a standard flow $f_\psi$ to learn $q_\sigma(\tilde{x})$ such that the corresponding density has the product form of equation (8), and the first $d-$components $u$ of the flow's output $f_\psi^{-1}(\tilde{x})$ are noise-insensitive whereas the remaining $(D-d)-$components $v$ remain noise-sensitive. Thus, intuitively, we want the first $d-$components to *denoise* the inflated data.

**DNF:** Let $f_\psi : \mathbb{R}^D \to \mathbb{R}^D$ be a standard flow with reference measure $p_z(z)$. We denote the first $d-$components of the flows output as $u$, and the remaining ones as $v$, i.e. $(u,v)^T = f_\psi^{-1}(\tilde{x})$. More formally,

$$u = u(\tilde{x}) = \text{Proj}_u(f_\psi^{-1}(\tilde{x})), \quad v = v(\tilde{x}) = \text{Proj}_v(f_\psi^{-1}(\tilde{x})) \tag{9}$$

with $\text{Proj}_u(z) = (z_1, \ldots, z_d)$ and $\text{Proj}_v(z) = (z_{d+1}, \ldots, z_D)$ for $z \in \mathbb{R}^D$. As reference measure $p_z$, we choose $p_z((u,v)^T) = p_u(u)p_v(v)$ with $p_v(v)$ modelling the noise-sensitive part, and $p_u(u)$ the noise-insensitive part. In particular , if $q_\sigma(\tilde{x}|x)$ is a $(D-d)-$dimensional Gaussian distribution with covariance $\sigma^2 I_{D-d}$, we set $p_v(v) = \mathcal{N}(v; 0, \sigma^2 I_{D-d})$.

For $p_u(u)$ to model the noise-insensitive part, we want the image $f_\psi(u, 0)$ to be in the manifold. Therefore, we embed $u$ back in $\mathbb{R}^D$ by padding the missing coordinates with 0,

$$\text{Pad}(u) = (u, \underbrace{0, \ldots, 0}_{(D-d)\text{-times}})^T, \tag{10}$$

such that the operation $\text{Pad}(\text{Proj}_u((u,v))) = (u, 0)^T$ ignores the noise-sensitive part $v$ in the latent space. This operator allows us to define a denoising function $r_\psi(\tilde{x})$ as

$$r_\psi(\tilde{x}) := (f_\psi \circ \text{Pad} \circ \text{Proj}_u \circ f_\psi^{-1})(\tilde{x}) \tag{11}$$

and we regularize $\psi$ by minimizing

$$\mathcal{C}(\psi) := \mathbb{E}_{x \sim p(x)} \mathbb{E}_{\tilde{x} \sim q_\sigma(\tilde{x}|x)} ||x - r_\psi(\tilde{x})||^2 \tag{12}$$

where $|| \cdot ||$ denotes the $L_2$ norm.

We have not specified the reference measure $p_u(u)$. To facilitate the disentanglement of noise-insensitivity and noise-sensitivity in the $u$ and $v$ variables, we transform $u$ with yet another flow $h_\phi$ with paramters $\phi$ and reference measure $p_{u'}$ (e.g. standard Normal).
Now, our sampling procedure looks as follows: 1. sample $u' \sim p_{u'}(u')$, 2. apply $h_\phi$ to obtain a sample $u$ from $p_u(u)$, i.e. $u = h_\phi(u')$, and 3. set $x = f_\psi(\text{Pad}(u))$. Denoting $\theta = (\psi, \phi)$, our model to learn $q_\sigma(\tilde{x})$ is

$$q_\theta(\tilde{x}) = |\det G_{f_\psi}(\tilde{x})|^{-\frac{1}{2}} p_u(u(\tilde{x})) p_v(v(\tilde{x}))$$
$$= |\det G_{f_\psi}(\tilde{x})|^{-\frac{1}{2}} |\det G_{h_\phi}(u(\tilde{x}))|^{-\frac{1}{2}} p_{u'}(h_\phi^{-1}(u(\tilde{x}))) p_v(v(\tilde{x})). \tag{13}$$

Note that the reconstruction loss, equation (12), is essentially the objective function for the *Denoising Autoencoder* introduced in [1]. Therefore, we call our method *Denoising Normalizing Flow* (DNF) and it is trained on

$$\mathcal{L}_{\text{DNF}}(\theta) := D_{\text{KL}}(q_\sigma(\tilde{x})||q_\theta(\tilde{x})) + \lambda \mathcal{C}(\psi) \tag{14}$$

where $\lambda > 0$ is the penalty hyperparameter and is trading the density estimation with the manifold learning. A graphical description of the DNF model is given in Figure 1 (a), and an algorithmic description in the DNF Algorithm below.[3]

**Answer to research question:** If $\mathcal{L}_{\text{DNF}}(\theta) = 0$, the reconstruction error expressed in equation (12) is 0. Thus, the generative story of the DNF is exactly the one described in point 1. of our research question with $g_\theta := f_\psi \circ \text{Pad} \circ h_\phi$. The inverse, $h_\phi^{-1} \circ \text{Proj}_u \circ f_\psi^{-1}$, can be used to infer $u'$ s.t. point 2. holds. Finally, to show the third claim, we additionally assume that $\mathcal{U}$ (which is the domain of the manifold generating function $g$) is diffeomorphic to $\mathbb{R}^d$, s.t. without loss of generality we set $\pi(u) = \mathcal{N}(u; 0, I_d)$. Then, we exploit equation (8) and calculate $|\det G_g(x)|$ efficiently with the help of $f_\psi$ and $h_\phi$, see Proposition 1 and its proof in the supplementary.

---

[3]We show the algorithm for the general scenario where the manifold is unknown and thus noise cannot be added to the normal space. We also ignore terms independent of $\theta$ in the calculation of $\mathcal{L}_{\text{DNF}}$, and denote this loss function as $\mathcal{L}'_{\text{DNF}}$.

**DNF Algorithm:** Training of Denoising Normalizing Flow for $q_\sigma(\tilde{x}|x) = \mathcal{N}(\tilde{x}; x, \sigma^2 I_D)$. For simplicity, we show a stochastic gradient descent with a constant learning rate. Alternative optimization methods and learning rate schedules can be easily adapted.

---

**Require:** Manifold dimension $d$, Learning rate $\alpha$, penalty parameter $\lambda$, inflation variance $\sigma^2$, batch size $n$, number of epochs $E$.
**Initialize:** Parameters $\psi$ and $\phi$ for flows $f_\psi$ and $h_\phi$.

   **while** $\theta = (\psi, \phi)$ has not converged **do**
      **for** $e = 1$ **to** $E$ **do**
         **for** $i = 1$ **to** $n$ **do**
            Sample: $x_i \sim p(x)$              # sample data
            Inflate: $\tilde{x}_i = x_i + \varepsilon_i$, where $\varepsilon_i \sim \mathcal{N}(0, \sigma^2 I_D)$     # add noise
            $(u_i, v_i) \leftarrow f_\psi^{-1}(\tilde{x}_i)$      # project on $u \in \mathbb{R}^d$ and $v \in \mathbb{R}^{D-d}$
            $\hat{x}_i \leftarrow f_\psi(u_i, 0)$           # reconstruct $x$
            $u'_i \leftarrow h_\phi^{-1}(u_i)$           # transform $u$
         **end for**
         $\mathcal{L}'_{\text{DNF}} \leftarrow \frac{1}{n} \sum_i^n \log p_{u'}(u'_i) - \log|\det J_{h_\phi}(u'_i)| + \log(p_v(v_i)) - \log|\det J_{f_\psi}(u_i, v_i)|$
               $+\lambda||x_i - \hat{x}_i||^2$      # calculate $\log q_\theta(\tilde{x})$ and add reconstruction error
         $\theta \leftarrow \theta - \alpha \nabla_\theta \mathcal{L}'_{\text{DNF}}$      # update model parameters
      **end for**
   **end while**

---

**Proposition 1** *Let $\mathcal{M}$ be a $d-$dimensional manifold embedded in $\mathbb{R}^D$ through $g : \mathbb{R}^d \to \mathcal{M}$. Let $x \sim p(x)$ be generated by $g(u)$, where $u \sim \mathcal{N}(u; 0, I_d)$, i.e. $x = g(u)$. Assume that we can learn the inflated distribution $q_\sigma(\tilde{x})$ using an NF and that for $\tilde{x} = x$ it holds that $q_\sigma(x) = p(x)q_\sigma(x|x)$. If $\mathcal{L}_{DNF}(\theta) = 0$, then*

$$|\det G_g(x)| = |\det G_{f_\psi}(x)||\det G_{h_\phi}(x)|. \tag{15}$$

## 4 Related work

**Autoencoders:** An autoencoder learns to compress the data $x$ using an encoder $f_\psi^{-1}$, and then to reconstruct $x$ using a decoder $g_\theta$. As the AE is only trained on the reconstruction error $||x - g_\phi(f_\psi^{-1}(x))||^2$, it can't be used to generate new data or estimate $p(x)$. However, if the input is corrupted by Gaussian noise $\varepsilon$, an optimal AE that can reconstruct the uncorrupted input depends on the gradient of the data-loglikelihood [1]. This can be exploited to estimate $p(x)$ [7]. A variational autoencoder (VAE) [26] is a stochastic version of an AE. More concretely, a lower bound on the data log-likelihood, known as ELBO or free energy, is maximized to learn parameters $\phi$ and $\theta$ such that the true conditional distributions $p(x|z)$ and $p(z|x)$ are approximated by $p_\theta(x|z) = \mathcal{N}(x; g_\theta(z), I_D)$ and $q_\psi(z|x) = \mathcal{N}(x; f_\psi(x), I_D)$, respectively. If one is not interested in $p(x)$ but still wants to generate new data using an AE, regularization terms can be exploited [28, 37]. Alternatively, an NF can be used to normalize the learned latent variables $u = f_\psi(x)$. Thus, after training a usual AE, the latent density associated with the latent variable $u$ is learned using a standard NF. This probabilistic autoencoder (PAE) was recently introduced in [9].

**NFs based models:** Recently, a few attempts have been made to overcome the topological constraint of NFs in terms of density estimation [10, 11, 20, 31, 34, 35], sampling [4, 10, 12, 24], and inference [5, 10]. If the manifold is known, i.e. its atlas is given, a usual flow in $\mathbb{R}^d$ is sufficient to learn the data-density $p(x)$ as we can use these charts to push-back $p(x)$ to $\mathbb{R}^d$. This was first done in [16] for a manifold consisting of a single chart $g$. However, if $g$ is not known, only recently some methods were proposed to learn it [4, 10]. We review these methods in the following, highlight their differences to the DNF and illustrate them in Figure 1.

**Pseudo invertible encoder (PIE):** The idea of [4] is to define the manifold as a level set of a usual NF $f$. For that, they propose to treat the first $d-$latent variables $u$ differently than the remaining $D - d$ variables $v$, by using different reference measures for $u$ and $v$, respectively, i.e. $f^{-1}(x) = z = (u, v)^T \in \mathbb{R}^d \times \mathbb{R}^{D-d}$ and $p_z(z) = p_u(u)p_v(v)$. The gist is very similar to the DNF. However, there is no incentive for $f$ to encode the manifold in $u$ which manifests in poor sample quality (see [10]). In addition, this approach does not work for data living exactly on a

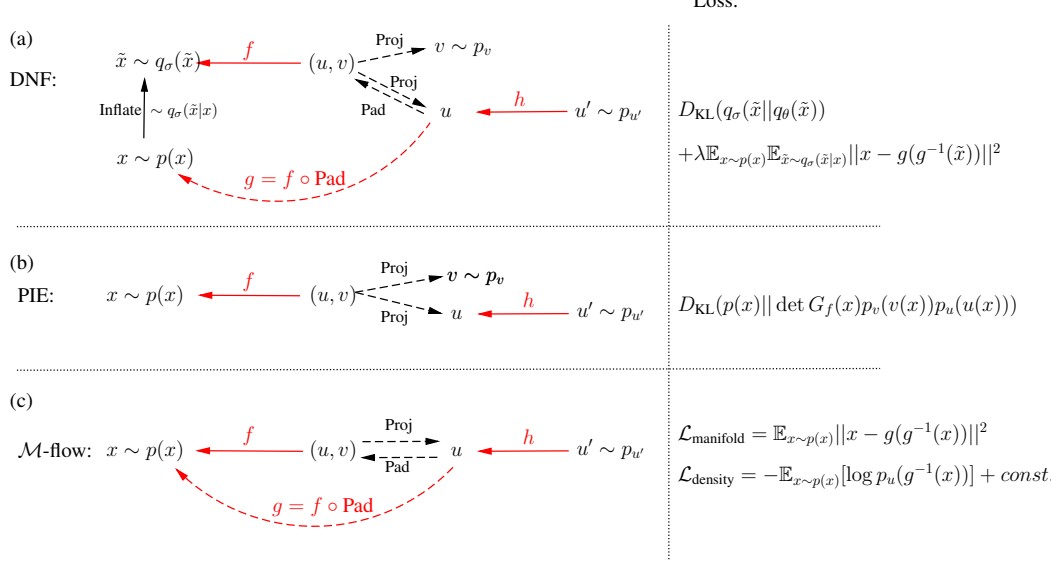

Figure 1: Schematic overview of different NF-based methods (DNF, PIE, $\mathcal{M}$-flow) (adopted from [10]). **Left:** Graphical model. Red solid lines are NFs, red dashed lines the corresponding generator. Black dashed lines describe the projection or padding operation, respectively, as described in the main text. **Right:** The losses used to train the models.

low-dimensional manifold as the Jacobian determinant of $f$ degenerates. In Figure 1, we depict a version of the PIE model where $u$ is additionally transformed with a flow $h$. Note that a PIE model is an DNF with $\lambda = \sigma = 0$.

**Manifold flow ($\mathcal{M}-$flow):** The idea of [10] is to learn the embedding $g$ directly by first mapping the data into the latent space using a usual NF $f$, projecting it to the first $d-$coordinates $u$, and finally transform these variables using yet another flow $h$ in $\mathbb{R}^d$ with reference measure $p_{u'}$, see Figure 1 for a depiction of their $\mathcal{M}-$flow. The resulting density,

$$p_{\mathcal{M}}(x) = p_{u'}(h^{-1}(u(x)))|\det G_h(u(x))|^{-\frac{1}{2}}|\det G_g(u(x))|^{-\frac{1}{2}}, \tag{16}$$

is indeed a density defined only on the manifold. As noted in [10], calculating the Gram determinant of $g$ may be very slow for large $d$. Therefore, they propose to train the parameters of $g$ using the reconstruction error $||x - g^{-1}(g(x))||_2^2$ only, and the parameters of $h$ using the usual $D_{\text{KL}}$ objective for NFs. They propose several training strategies to ensure that $g$ encodes the manifold (manifold phase), and $h$ learns the density (density phase). One strategy alternates between the manifold- and density phase (alternate training) for every training epoch. Another strategy first trains $g$ on the reconstruction error and then learns the density through $h$ (sequential training). Note that a PAE is also trained using a sequential training scheme, as opposed to the DNF which combines the manifold- and density learning phase into a single cost function, equation (14). Another difference to the DNF is that the Gram determinant of $f$ is neither used to train the $\mathcal{M}-$flow nor to estimate $p(x)$.

## 5   Results

In this chapter, we confirm experimentally our claims made at the end of Section 1, formalized as a research question, and answered at the end of Section 3. First, we show that the DNF circumvents the degeneracy problem of a standard NF for manifold-supported densities. Afterward, we show that the DNF can learn meaningful latent representations for naturalistic images, without sacrificing the sample quality compared to the $\mathcal{M}-$flow, PAE, and VAE (see Section 4). Our method depends on two hyperparameters, the noise magnitude $\sigma^2$ and the penalty coefficient $\lambda$. $\sigma^2$ needs to be sufficiently small such that equation (8) approximately holds (see [20] for more details on the choice of $\sigma$). If $\lambda = 0$, only the density is learned, and if $\lambda \gg 1$ the manifold learning dominates.

We closely follow the experimental setting of [10]. For the $\mathcal{M}-$flow and DNF, we use the same network architectures and training protocols. These architectures are based on affine-coupling layers [13], neural splines [14], and trainable permutations [25]. The PAE uses convolutional neural networks for the encoder and decoder, respectively. For images, we use a recently developed variant of the VAE, the InfoMax-VAE [33], which uses insights from information theory to learn meaningful latent representations. We refer to the supplementary for more training details. [4]

## 5.1 Density estimation

We consider a $1-$dimensional manifold embedded in $\mathbb{R}^2$, a thin spiral. We draw the latent variable $u$ from an exponential distribution with scale 0.3, i.e. $\pi(u) \propto \exp(-0.3u)$, and generate the spiral through $g(u) = \frac{\alpha\sqrt{u}}{3}(\cos(\alpha\sqrt{u}), \sin(\alpha\sqrt{u}))^T$ where $\alpha = 4\pi/3$ (upper half of Figure 2 top left).

We train a standard NF *(upper right)*, $\mathcal{M}-$flow *(middle left)*, DNF *(middle right)*, PAE *(lower left)*, VAE *(lower right)* with similar architectures on 100 epochs with a batch size of 100. As we can see in Figure 2, the Jacobian determinant of the standard NF degenerates, and the learned density collapses into single points. Surprisingly, the $\mathcal{M}-$flow fails to learn $p(x)$ as well, no matter which training schedule we use (we display the result of the sequential training). The PAE learns an inflated version of $p(x)$, however, it does not have a deflation procedure. The standard VAE simply equates $p(x)$ with the Gaussian prior on the latent variables (we take a point estimate of the ELBO to approximate $p(x)$).

For the DNF, we use Gaussian noise with $\sigma^2 = 0.01$ and $\lambda = 1$. Note that only the DNF is trained on the inflated density $q_\sigma(\tilde{x})$, and it encodes the noisy part in $v(\tilde{x})$, i.e. for points close to the manifold $v(\tilde{x})$ should be close to 0. Therefore, we set $p(x)$ to 0 whenever $v(x) > \sigma/3$ and otherwise we approximate $p(x)$ according to equation (8) with $q_\theta(x)/q_\sigma(x|x)$ where $q_\sigma(x|x) = 1/\sqrt{2\pi\sigma^2}$. The fact that the $\mathcal{M}-$flow fails to learn $p(x)$ and the DNF succeeds, suggests the regulatory importance of the Gram determinant of $f_\psi$ (which is only used from the DNF, and reflects the main difference between those two methods next to the training scheme). When setting $\sigma^2 = 0$, the DNF degenerates similar to the standard NF (more details in the supplementary). This illustrates the importance of the inflation step for a density supported on a low-dimensional manifold.

All densities are evaluated on a $100 \times 100$ grid. For that, the $\mathcal{M}-$flow calculates the Gram determinant of the learned generator $g$ (see equation (16) and Figure 1 (c)) for each point. Like this, evaluating the density on a batch consisting of 500 points takes about 188 seconds on a GPU[5]. On the same device and for the same batch size, the DNF needs only 1 second to evaluate the density. This illustrates the drawback of needing to calculate the full Gram determinant (see Section 2).

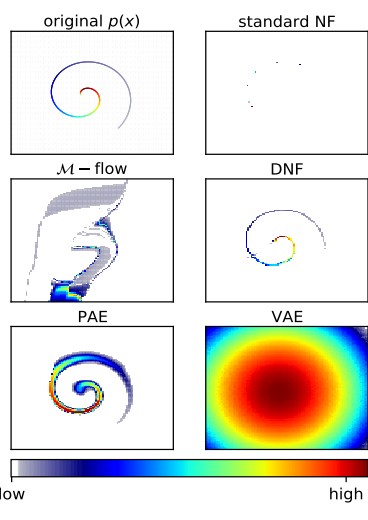

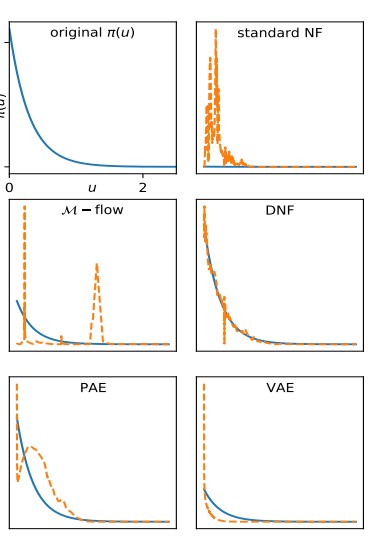

Figure 2: Density estimation

---

[4]Our main code is available at `https://github.com/chrvt/denoising-normalizing-flow` and is based on the original $\mathcal{M}-$flow implementation made public by the authors of [10] under the MIT license. For the InfoMax-VAE and PAE on images, we use the official implementations made public by the authors of [33] and [9] under the Apache License 2.0 and General Public License v3.0, respectively.

[5]The Gram matrix can be calculated using automatic differentiation.

|  | StyleGAN $d = 2$ | | StyleGAN $d = 64$ | |
| --- | --- | --- | --- | --- |
| Model | FID | Mean reconstr. error | FID | Mean reconstr. error |
| $\mathcal{M}-$flow | $4.85 \pm 0.14$ | $309.32 \pm 6$ | $19.95 \pm 0.22$ | $1019.18 \pm 30.08$ |
| DNF | $\mathbf{4.42} \pm 0.2$ | $\mathbf{225.78} \pm 14.4$ | $\mathbf{17.61} \pm 0.11$ | $\mathbf{899.35} \pm 37.19$ |
| InfoMax-VAE | $38.2 \pm 1.19$ | $12047.92 \pm 105.95$ | $58.18 \pm 6.67$ | $16397.54 \pm 286.65$ |
| PAE | $22.58 \pm 4.50$ | $7181.21 \pm 27.17$ | $54.51 \pm 7.13$ | $7169.97 \pm 14.25$ |

Table 1: FID and mean reconstruction error of the $\mathcal{M}-$flow, DNF, InfoMax-VAE, and PAE on the StyleGAN image manifold for $d = 2$ and $d = 64$. For $d = 2$, we train 10 models with different initializations, remove the best and worst results, and report the mean and standard deviation of the remaining 8 models. For $d = 64$, we train 3 models and report the mean and standard deviation.

To further validate the learned on-manifold density, for each model we compare the induced latent density with the ground truth exponential distribution $\pi(u)$ (second half of Figure 2). For that, we first generate 1000 latent points $u \sim \pi(u)$, then evaluate the model likelihood at $x = g(u)$, and finally multiply the likelihood with the square root of the Gram determinant of the generating mapping $g$, $\sqrt{\det G_g(u)} \propto (1 + (\alpha u)^2)/(\alpha u)^2$. This must lead to the correct latent density $\pi(u)$ if the data-density $p(x) = |\det G_g(x)|^{-\frac{1}{2}} \pi(g^{-1}(x))$ is learned correctly as $p(x)$ is uniquely determined by the pair $(g, \pi)$. The standard NF, $\mathcal{M}-$flow and VAE are out of scale. The PAE follows the exponential course of $\pi(u)$, however, only the DNF learns the correct course and scale. This indicates that equation (15) can be used to approximate $\det G_g(u)$ well.

In the supplementary, we conduct more density estimation experiments and compare the DNF with the inflation-deflation method [20]. For a von Mises distribution on a circle, and a mixture of von Mises distributions on a sphere, the DNF learns the density almost exactly showing the correctness of equation (15). Also in the supplementary, we use the DNF for probabilistic inference following the protocol used in [10].

## 5.2 StyleGAN image manifold

One drawback of both the $\mathcal{M}-$flow and DNF is that the true manifold dimension $d$ must be known beforehand. For real-world datasets, $d$ is unknown. Therefore, [10] uses a StyleGAN2 model [23] trained on the FFHQ dataset [22] to generate an $d-$dimensional manifold by only varying the first $d$ latent variables while keeping the remaining fixed.

$\mathbf{d = 2}$: We want to show that the DNF learns meaningful latent representations. For that, we first train an DNF on $10^4$ images using 100 epochs with $\sigma^2 = 0.1$ and $\lambda = 1000$. Then, we generate an image grid by varying the $2-$dimensional latent variables $u'$ and mapping them to the data space using the learned mapping $g_\theta$. In Figure 3, we see on the top left the original grid obtained by the StyleGAN2 model, and on the top right the grid generated by the DNF. A smooth mapping from latent to data space is learned. The PAE (lower left), does not learn such a smooth mapping which may be due to the lack of learning $p(x)$, in contrast to the InfoMax-VAE (lower right). However, the latter lacks variability which shows that the learned posterior $p(z|x)$ does not match the prior $\mathcal{N}(z; 0, I_d)$.

To further evaluate the quality of generated test samples, we display the Fréchet inception distance (FID score) [6] [19, 30] in Table 1 along with the mean reconstruction error which measures the manifold learning. We slightly outperform the $\mathcal{M}-$flow in terms of FID score, and significantly in terms of the mean reconstruction error. The latter is surprising as the $\mathcal{M}-$flow is directly trained on the reconstruction error. The PAE and InfoMax-VAE perform worse compared to the $\mathcal{M}-$flow and DNF indicating a suboptimal choice for the latent dimension for these models.

---

[6]The FID is the Wasserstein-2 distance between two Gaussians. The mean values and covariance matrices are obtained as sample estimates of the original and model data, respectively. However, instead of using the pixel values, one uses the outcomes of the next to the last layer of the Inception v3 ([36]) image classifier trained on the corresponding data set.

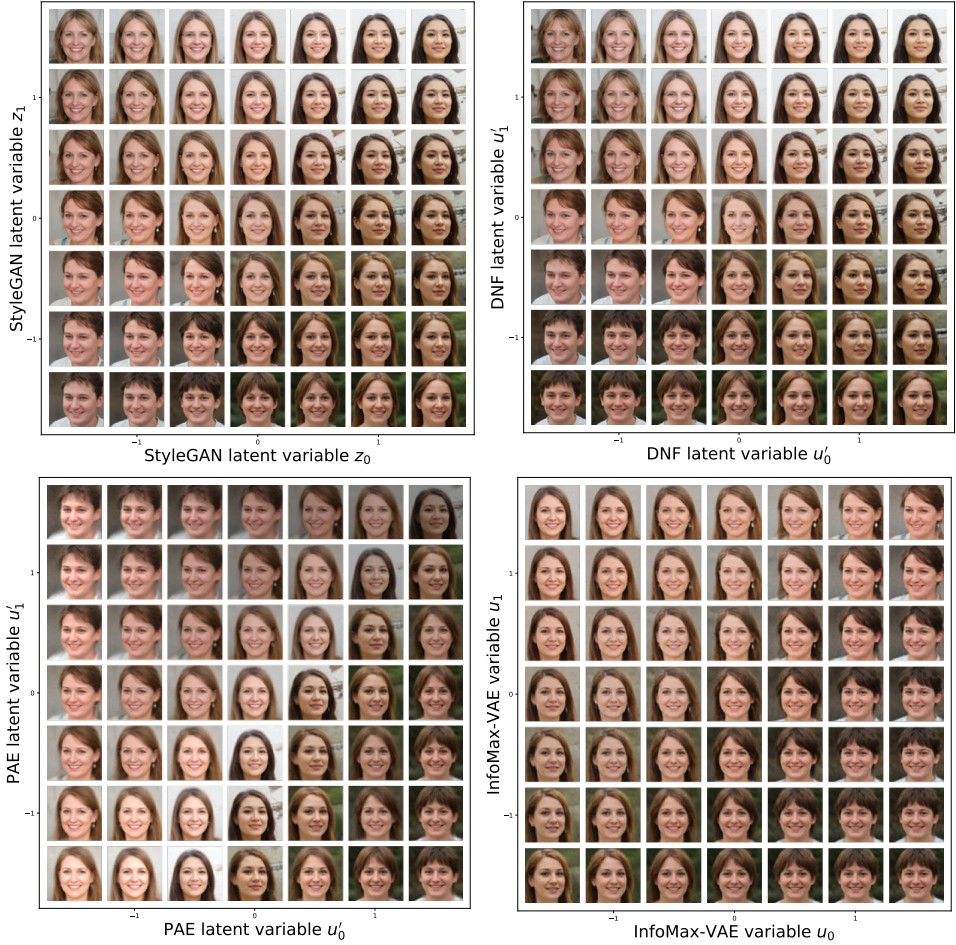

Figure 3: Image grids, as described in the main text. StyleGAN images (top left) are used to train an DNF (top right), PAE (bottom left), and InfoMax-VAE (bottom right).

$\mathbf{d = 64}$: We train the models on $2 \cdot 10^4$ images for 200 epochs. In Figure 4, we show in the first 5 columns samples from the original dataset (top), $\mathcal{M}-$flow (second row), DNF (third row), InfoMax-VAE (fourth row), and PAE (last row). In the remaining 5 columns, we show how smoothly these models interpolate linearly in the latent space. For that, we linearly interpolate between two training images in latent space, and display the corresponding trajectory in image space. We compare the FID and mean reconstruction error of the models in Table 1. Similar to the $d = 2$ case, the DNF outperforms the $\mathcal{M}-$flow, InfoMax-VAE, and PAE in terms of FID and mean reconstruction error.

## 6 Summary and discussion

Our model is based on the theoretical work established by [20] leading to a natural combination of NFs and DAE - the DNF. In contrast to similar methods, an DNF is trained on a single objective function combining manifold- and density learning. To learn a density supported on a low-dimensional manifold with an NF, one needs to compute the flow's Gram determinant. The DNF circumvents this necessity and can be used to approximate it. We have pigeonholed the DNF into the literature, and compared its performance on naturalistic images with related methods ($\mathcal{M}-$ flow, PAE, VAE). Among those methods, we have seen that the DNF generates images with the highest quality (in terms of FID), and reconstructs a given input with the lowest $L_2$ distance.

It is well known that adding noise in the input can increase the generalisation performance in supervised learning tasks [2, 8, 29]. However, typically this comes with the price of lower sample quality or worse density estimation. The DNF has the potential to avoid this trade-off, and proves

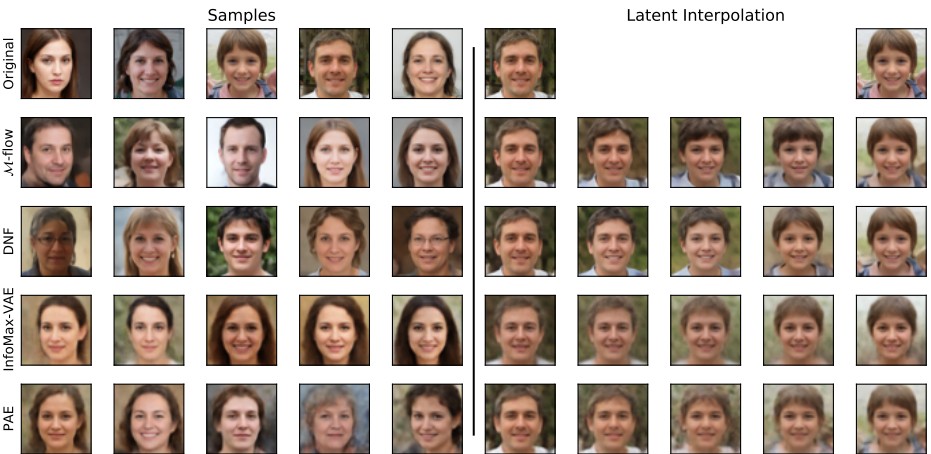

Figure 4: **First 5 columns:** Samples from StyleGAN, $\mathcal{M}-$flow, DNF, InfoMax-VAE, and PAE (in this order). **Last 5 columns:** Latent interpolation as described in the main text.

experimentally that adding noise to images can lead to better performance in terms of sampling quality and manifold learning.

**Extensions:** The theoretical foundation of the DNF is equation (8) which requires the manifold to be sufficiently smooth, and the noise to be added in the manifold's normal space. Although for the special case where $d \ll D$ standard Gaussian noise is an excellent approximation for a Gaussian in the normal space, more research is needed on how to sample in a manifold's normal space in order to improve the performance of the DNF. Essentially, the DNF learns to encode the manifold in the $u$ component and the normal space direction in the $v$ component. Arguably, a pre-trained DNF could be used to improve the sampling ability in the normal space.

Another limitation of the DNF is that the manifold dimension $d$ needs to be known. Nevertheless, if $p(x)$ is supported on a low-dimensional manifold, to the best of our knowledge the DNF together with the $\mathcal{M}-$flow are the only existing methods to learn $p(x)$ and the manifold generating mapping based on NF. Hence, in contrast to other methods which need to know the exact manifold, the DNF and $\mathcal{M}-$flow can still be used when $d$ is estimated from samples. The latter is an active research area [5, 15, 18].

Other recently developed methods (such as PAE or $\mathcal{M}-$flow) separate the manifold from the density learning. We have seen that this separation is not necessary, and have used different heuristics to evaluate the quality of learning. For the value of $p(x)$, we visualized the learned (latent) density (Figure 2). For the quality of latent representations, we generated an image grid (Figure 3), and for the smoothness in the latent space, we generated an image path (Figure 4). We measured the sampling quality using the FID, and the manifold-learning using the mean reconstruction error (Table 1). Given these different heuristics, there is an aspiration for a unified performance criterion.

**Broader Impact:** To compress increasingly high-dimensional data with the least loss of information as possible is becoming increasingly important. The DNF ties in with the $\mathcal{M}-$flow and shows that such compression is possible, even for naturalistic images. Similar to all other generative models, a possible negative impact of DNFs would be the generation of fake data. On a more positive note, the DNF could improve out-of-distribution detection or even increase the robustness to adversarial attacks which are both essential for reliable societal applications of Machine Learning.

## Acknowledgments and Disclosure of Funding

We thank Camille Gontier, Gagan Narula, Hui-An Shen, Katharina A. Wilmers, Nicolas Zucchet, and the anonymous reviewers for helpful comments. This study has been supported by the Swiss National Science Foundation grant 31003A_175644.

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
