## A    Proof of Proposition 1

Note that $p(x)$ is uniquely determined by the pair $(\pi, g)$, and that we set without loss of generality $\pi(u') = \mathcal{N}(u'; 0, I_d)$ as we assumed that $\mathcal{U}$ is diffeomorphic to $\mathbb{R}^d$. Since $\mathcal{L}_{\text{DNF}}(\theta) = 0$, we have that $x = f_\psi(\text{Pad}(h_\phi(u')))$ is distributed according to $p(x)$ when $u' \sim \mathcal{N}(u'; 0, I_d)$, and thus it must hold that

$$(f_\psi \circ \text{Pad} \circ h_\phi)(u') = g(u'), \tag{1}$$

for all $u' \in \mathbb{R}^d$. Therefore,

$$(h_\phi^{-1} \circ \text{Proj}_u \circ f_\psi^{-1})(x) = g^{-1}(x), \tag{2}$$

for all $x \in \mathbb{R}^D$. Note that $u(x) = (\text{Proj}_u \circ f_\psi^{-1})(x)$ so that the above equation can be written as

$$(h_\phi^{-1}(u(x)) = g^{-1}(x). \tag{3}$$

By assumption, we have that $D_{\text{KL}}(q_\sigma(\tilde{x})||q_\theta(\tilde{x})) = 0$ and $q_\sigma(x) = p(x)q_v(x|x)$. Thus,

$$p(x)q_\sigma(x|x) = q_\theta(x)$$

$$p(x) = |\det G_{f_\psi}(x)|^{-\frac{1}{2}} |\det G_{h_\phi}(x)|^{-\frac{1}{2}} \left( \frac{p_{u'}(h_\phi^{-1}(u(x)))p_v(v(x))}{q_\sigma(x|x)} \right)$$

$$|\det G_g(x)|^{-\frac{1}{2}} \pi(g^{-1}(x)) = |\det G_{f_\psi}(x)|^{-\frac{1}{2}} |\det G_{h_\phi}(x)|^{-\frac{1}{2}} \left( \frac{p_{u'}(h_\phi^{-1}(u(x)))p_v(v(x))}{q_\sigma(x|x)} \right)$$

$$|\det G_g(x)|^{-\frac{1}{2}} = |\det G_{f_\psi}(x)|^{-\frac{1}{2}} |\det G_{h_\phi}(x)|^{-\frac{1}{2}} \left( \frac{p_{u'}(h_\phi^{-1}(u(x)))p_v(v(x))}{\pi(g^{-1}(x))q_\sigma(x|x)} \right) \tag{4}$$

where for the second equation we have used the definition of $q_\theta$ (equation (13) in the main text), and for the third equation the definition of $p(x)$. Since $p_{u'}(u') = \pi(u') = \mathcal{N}(u'; 0, I_d)$ and equation (3) holds, we have that

$$p_{u'}(h_\phi^{-1}(u(x))) = \pi(g^{-1}(x)). \tag{5}$$

Note that we can choose $p_v$ to match $q_\sigma$ such that $p_v(0) = q_\sigma(x|x)$. For instance, if $q_\sigma$ is a Normal distributions with covariance matrix $\sigma^2 I_d$, we set $p_v(v) = \mathcal{N}(v; 0, \sigma^2 I_d)$. In any case, $v(x)$ must be 0 as otherwise $f_\psi$ would not bijective. Therefore, equation (4) yields

$$|\det G_g(x)| = |\det G_{f_\psi}(x)| \cdot |\det G_{h_\phi}(x)| \tag{6}$$

which was to be shown.

## B    Experiments

We first describe the main ingredients of the architectures used to construct the flows $f_\psi$ and $h_\phi$. These architectures are based on (multi-scale) coupling layers ([3]), neural splines ([4]), and trainable permutations ([10]).

**Coupling layers:** Given an input $x = (x_1, \ldots, x_N)^T \in \mathbb{R}^N$, a coupling layer is a bijective transformation of $x$. As mentioned in the main text, for an NF to be tractable, the Jacobian of this transformation should be efficiently computable. For instance, an affine-coupling ([3]) splits $x$ into two halves, $x_{1:n} = (x_1, \ldots, x_n)^T$ and $x_{n+1:N} = (x_{n+1}, \ldots, x_N)^T$, leaves the first half $x_{1:n}$ unchanged, and the second half is affine-linearly transformed. The parameters of this affine-linear transformation are outputs of a neural network with input $x_{1:n}$. Thus, denoting the output of this transformation as $y$, we have that

$$y_{1:n} = x_{1:n}$$
$$y_{n+1:N} = x_{n+1:N} \odot \exp(s(x_{1:n})) + t(x_{1:n}), \tag{7}$$

where $s$ and $t$ are neural networks mapping $\mathbb{R}^n$ to $\mathbb{R}^{N-n}$, and $\odot$ is the element-wise product. The Jacobian matrix of this transformation is triangular and the log-determinant is given by $\sum_{k=1}^{N-n} s_k(x_{1:n})$.

**Masking:** With a single coupling layer, the first $n$ components remain unchanged. However, several coupling layers can be composed where for each new layer, a different subset of components stays unchanged and the remaining ones are transformed. For vector data, a checkerboard pattern (i.e. a fixed permutation) was proposed in [3], and for images, an additional channel-wise masking. Additionally,a linear, LU-decomposed transformation can be applied. Also, the batch can be normalized before passing it to the coupling transformation. The composition of batch normalization, coupling layer, and masking operation is compactly denoted as a flow step.

**Glow:** In [10], the fixed permutation was replaced by a learnable one. For images, this corresponds to an invertible $1 \times 1$ convolution. Additionally, [10] proposed to replace the batch normalization with a learnable affine transformation per channel - an ActNorm layer. This aims to alleviate the problem of training large networks with small minibatches, see [10] for more details. Thus, a flow step in Glow consists of applying an actnorm layer, a $1 \times 1$ convolution, and the coupling transformation.

**Multi-scale:** To overcome the computational burden for high-resolution images, a multi-scale architecture was proposed in [3]. Before applying a series of flow steps, a squeezing operation reshapes the input tensor to reduce the spatial resolution while increasing the channels. After applying the series of flow steps, half of the variables are directly modeled as latent variables and the other half serves as input for the next scale.

**Neural splines:** To allow for more flexible coupling transformations, [4] proposed to replace the affine-linear with a monotonic rational-quadratic mapping. More precisely, an interval $[-B, B]$ is split into $K$ equidistant bins, and on each subinterval, a rational-quadratic spline is defined such that the derivatives are continuous at the boundary points. The parameters of the splines are again outcomes of neural networks. We refer to $B$ as the spline range and $K$ as the bin size in the following. Outside of the interval $[-B, B]$, the transformation is set to the identity.

## B.1  Density estimation

**Dataset:** The thin spiral is a 1-dimensional manifold embedded in $\mathbb{R}^2$. We generate a training set by sampling $10^4$ Exp(0.3)-distributed latent variables $u'$, setting $u = \sqrt{u'} \cdot 540 \cdot (2\pi)/360$ and mapping $u$ to the data space by applying $g(u) = u(\cos(u), \sin(u))^T$. Out of those $10^4$ samples, we reserve 10% as validation set.

**Training protocol:** We train on 100 epochs with a batch size of 100 (i.e. in total 9000 gradient steps), and take the model yielding the best result on the validation set. We use the AdamW ([12]) optimizer with annealing of the learning rate to 0 after 100 epochs using a cosine schedule ([11]).

**Architectures:** The $\mathcal{M}-$flow and DNF consist of two NFs, $f_\psi$ and $h_\phi$. The standard NF consists only of $f_\psi$. For all models, we use rational-quadratic splines as coupling layers for $f_\psi$ with trainable, LU-decomposed linear transformation without batch normalization or dropout (as mentioned in [2], these operations introduce stochasticity leading to problems with the invertibility of $f_\psi$ and $h_\phi$). For the flow $h_\phi$, we use only one coupling layer, an affine-autoregressive transformation introduced in [13], as the latent dimension is 1, see Table S1. The parameters for the coupling transforms are obtained from a residual network with two residual blocks of 100 units in each layer and ReLU activation.

| model | # levels | flow $f_\psi$ | | | flow $h_\phi$ | | |
|---|---|---|---|---|---|---|---|
| | | # couplings | coupling type | masking | # couplings | coupling type | masking |
| standard NF | 1 | 10 | spline with $B = 3, K = 5$ | LU | - | - | - |
| $\mathcal{M}$-flow & DNF | 1 | 9 | spline with $B = 3, K = 5$ | LU | 1 | affine-autoregressive | - |

Table S1: Architectures for standard NF, $\mathcal{M}-$flow, and DNF on the thin spiral dataset

The PAE and VAE consist both of an encoder and decoder network. We use a residual network for both encoder and decoder with 5 blocks for the PAE and 10 blocks for the VAE. Each block has 100 hidden units with ReLU activation. After having trained the encoder and decoder of the PAE on the mean squared reconstruction error, the latent variables are transformed using the same architecture for $h_\phi$ as in the $\mathcal{M}-$flow and DNF, respectively.

In Table S2, we report the total number of parameters used in each model.

|  | standard NF | $\mathcal{M}-$flow | DNF | PAE | VAE |
|---|---|---|---|---|---|
| # parameters in million | 0.4 | 0.4 | 0.4 | 0.4 | 0.46 |

Table S2: Total number of parameters (in million) used to learn the density on the thin spiral by the various models.

**Density evaluation:** For the standard NF, DNF, and $\mathcal{M}-$flow, the density evaluation is straightforward. For the DNF, we additionally constrain the $v-$component to be small, as described in the main text. The PAE has no intrinsic estimation for $p(x)$, but estimates the latent distribution $\pi(z)$. To estimate $p(x)$, [1] marginalize over the latent space, i.e. $p(x) = \int_z \hat{\pi}(z)p(x|z)dz$, where for $p(x|z)$ a Gaussian ansatz is chosen. To approximate this intractable integral, a Laplace ansatz followed by a maximum posterior (MAP) estimate for $z$ is made (see [1] for more details).

The VAE with encoder $f_{\phi_1}$, variance $\sigma_{\phi_2}$, and decoder $g_\theta$, is trained on the following lower bound on $p(x)$:

$$\log p(x) \geq \mathcal{L}_{\text{VAE}}(\phi, \theta; x) := -D_{\text{KL}}[q_\phi(z|x)||\pi(z)] + \mathbb{E}_{z \sim q_\phi(z|x)}[\log p_\theta(x|z)], \tag{8}$$

where $p_\theta(x|z) = \mathcal{N}(x; g_\theta(z), I_D)$ and $q_\phi(z|x) = \mathcal{N}(x; f_{\phi_1}(x), \sigma_{\phi_2}(x)I_D)$. We use an estimate of this lower bound to approximate $p(x)$. For that, we approximate the second term using a Monte-Carlo estimate for the integral over $q_\phi(z|x)$ by sampling one $z \sim q_\phi(z|x)$ and calculating $\log p_\theta(x|z)$.

## B.2 StyleGAN image manifold

**Datasets:** The StyleGAN image manifolds were introduced in [2]. A pre-trained StyleGAN network ([8]) on the FFHQ dataset ([7]) consisting of 512 latent variables $z$ and additional noise variables was used to generate a $d$-dimensional image manifold as follows: 1) generate a single sample $z$ from a standard Gaussian (and another one for the noise variables with smaller variance), 2) vary the first $d$ latent variables while keeping the others fixed.

**Preprocessing:** All 8-bit images are downsampled to a resolution of $64 \times 64$ and are preprocessed through uniform dequantization as in [10].

**Regularization:** For the residual networks generating the parameters for the coupling transformations, we apply weight decay with a prefactor of $10^{-6}$ without dropout. Furthermore, for the $\mathcal{M}-$flow and DNF, a $L_2-$regularization on the latent variables $u$ and $v$ with a prefactor of $0.01$ was used to stabilize the training.

**Metric:** As in [2], we use the PyTorch implementation [15] to calculate the FID scores.

### B.2.1 StyleGAN $d = 2$

**Dataset:** The dimensionality of the manifold is $d = 2$ and it is embedded in $\mathbb{R}^D$ with $D = 64 \cdot 64 \cdot 3 = 12288$. For training, $10^4$ images were generated of which 10% are used for validation, and $10^3$ images were generated for testing purposes.

**Training protocols:**

$\mathcal{M}-$*flow, DNF:* We train on 100 epochs with a batch size of 25 (i.e. in total 36000 gradient steps), and take the model yielding the best result on the validation set. We use AdamW optimizer ([12]) and anneal the learning rate to 0 after 100 epochs using a cosine schedule ([11]).

*InfoMax VAE:* We follow the original implementation of the InfoMax VAE on the CelebA dataset which uses Adam optimizer ([9]) and does not use a learning rate schedule. To avoid numerical instabilities, we use a batch size of 100 as recommended by the authors of [14].

*PAE:* We follow the original implementation of the PAE trained on the CelebA dataset setting the latent dimension to 2.

**Architectures:**

$\mathcal{M}-$*flow and DNF:* We use the same architecture proposed originally in [2]. Thus, a multi-scale architecture where each scale applies several rational-quadratic coupling transformations using the

Glow setting (i.e. with an ActNorm and $1 \times 1$ convolution layer) after a squeezing transformation, see Table S3. After the multi-scale transformation, [2] transformed the output with an invertible (LU-decomposed) layer, an invertible activation function, and another invertible layer acting on the first few channels per scale. The reason for this post-processing is to give the model some flexibility to align the manifold with features across different scales (see [2]).

The parameters for the rational-quadratic splines are obtained from a residual network with two residual blocks of 100 units in each layer and ReLU activation.

| | | flow $f_\psi$ | | | flow $h_\phi$ | | |
|---|---|---|---|---|---|---|---|
| model | # levels | # couplings | coupling type | masking | # couplings | coupling type | masking |
| $\mathcal{M}$-flow/ DNF | 4 | 20 | spline with $B = 10, K = 11$ | glow | 6 | spline with $B = 10, K = 11$ | glow |

Table S3: Architectures for $\mathcal{M}-$flow and DNF on the StyleGAN $n = 2$ manifold.

*InfoMax VAE:* The encoder and decoder are based on convolutional neural networks, see [14] for more details. In addition to that, a good latent representation is enforced by maximizing the mutual information between latent and input variables. This mutual information is estimated using a multilayer perceptron (MLP) mapping an input of dimension $D + d$ to a scalar. For this MLP, 5 linear networks are used with leaky ReLU activation.

*PAE:* The encoder and decoder are based on convolutional neural networks. For the NF a realNVP [3] architecture with random permutations is chosen, see [1] for more details.

In Table S4, we report the total number of parameters used in each model.

| | $\mathcal{M}-$flow | DNF | PAE | InfoMax VAE |
|---|---|---|---|---|
| # parameters in million | 16.4 | 16.4 | 67.5 | 28.6 |

Table S4: Total number of parameters (in million) used to learn the density on the StyleGAN $d = 2$ dataset by the various models.

## B.3 StyleGAN $d = 64$

**Dataset:** The dimensionality of the manifold is $d = 64$ and it is embedded in $\mathbb{R}^D$ with $D = 64 \cdot 64 \cdot 3 = 12288$. For training, $2 \cdot 10^4$ images were generated of which 10% are used for validation, and $10^3$ images were generated for testing purposes.

**Training protocols:**

*$\mathcal{M}-$flow and DNF:* We train on 200 epochs with a batch size of 25 (i.e. in total $1.44 \cdot 10^5$ gradient steps), and take the model yielding the best result on the validation set. We use AdamW optimizer ([12]) and anneal the learning rate to 0 after 100 epochs using a cosine schedule ([11]).

*InfoMax VAE:* We follow the original implementation of the InfoMax VAE on the CelebA dataset which uses Adam optimizer ([9]) and does not use a learning rate schedule. Also, to avoid numerical instabilities, we used a batch size of 100 as recommended by the authors of [14].

*PAE:* We follow the original implementation of the PAE trained on the CelebA dataset setting the latent dimension to 64.

**Architectures:**

*$\mathcal{M}-$flow and DNF:* We use the same architecture proposed originally by [2]. It differs from the architectures used for the StyleGAN $d = 2$ manifold in two ways. First, 2 additional coupling transformations are used for $h_\phi$, i.e. 8 in total. Second, the number of channels per scale on which the post-processing is acting is doubled.
The parameters for the rational-quadratic splines are obtained from a residual network with two residual blocks of 100 units in each layer and ReLU activation.

*InfoMax VAE:* We use the same architectures as for $d = 2$.

*PAE:* The encoder and decoder are based on convolutional neural networks as for $d = 2$. To incorporate the higher dimensionality of the latent space, the NF is now based on neural splines and trainable permutations (glow), see [1] for more details.

In Table S5, we report the total number of parameters used in each model.

|  | $\mathcal{M}$−flow | DNF | PAE | InfoMax VAE |
|---|---|---|---|---|
| # parameters in million | 39.5 | 39.5 | 67.9 | 28.9 |

Table S5: Total number of parameters (in million) used to learn the density on the StyleGAN $d = 64$ dataset by the various models.

## C   Additional Experiments

In this section, we first investigate the role of $\sigma^2$ in the DNF. Then, we conduct further density estimation experiments. Finally, we apply the DNF to real-world data.

### C.1   Thin spiral and StyleGAN $2d$ with $\sigma^2 = 0$

How does the performance of the DNF depend on $\sigma^2$? Certainly, if $\sigma^2$ is too large, the data-manifold is too disturbed to be restored. For instance, if the data lives on a circle with radius $r$, setting $\sigma > r$ will garble the manifold substantially, and the circle can't be retrieved. In [6], this intuition was formalized and an upper bound (depending on the manifold's curvature) for $\sigma^2$ was derived. However, what if $\sigma^2$ is too small?

In Figure S1, we show the learned density on the thin spiral when using the DNF with $\sigma = 0$. Similar to the NF, the density degenerates as the Gram determinant of the flow $f_\psi$ degenerates.

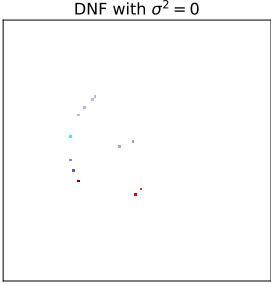

Figure S1: DNF with $\sigma^2 = 0$ on the thin spiral.

In Table S6, we show the FID and reconstruction error obtained on the StyleGAN $d = 2$ dataset using the DNF with $\sigma^2 = 0$ (together with the value for $\sigma^2$ used in the paper for comparison).

|  | StyleGAN $d = 2$ | |
|---|---|---|
| DNF with | FID | Mean reconstr. error |
| $\sigma^2 = 0.1$ | $4.42 \pm 0.2$ | $225.78 \pm 14.4$ |
| $\sigma^2 = 0.0$ | $4.38 \pm 0.26$ | $224.2 \pm 20.6$ |

Table S6: FID and mean reconstruction error of the DNF with different values of $\sigma^2$ on the StyleGAN image manifold for $d = 2$. We train 10 models with different initializations, remove the best and the worst result, and report the mean and standard deviation of the remaining 8 models.

There is no significant difference in performance. There might be two reasons for that. First, the dequantization in the pre-processing is essentially a noise inflation preventing the DNF to degenerate. Second, the StyleGan image manifold is not a manifold in a strict mathematical sense as it is generated by a GAN and not an immersion.

In summary, inflation seems to be crucial only for densities strictly supported on low-dimensional manifolds.

## C.2 Density estimation on a circle and sphere

**Dataset:** In [6], a von Mises distribution on a circle, and a mixture of von Mises distributions on a sphere was learned using the proposed inflation-deflation method.

*Circle:* Given samples from $\pi(z) \propto \exp(8\cos(z))$, the dataset consists of points generated by the mapping $g(z) = 3(\cos(z), \sin(z))$. Thus, the Gram determinan of $g$ is $\det G_g = 3$. We show samples from the induced distribution $p(x)$ in Figure S2 (top right).

*Sphere:* The latent distribution is a mixture of four von Mises distributions, $p_1^*(\phi_1, \theta_1), p_2^*(\phi_2, \theta_2), p_3^*(\phi_3, \theta_3)$ and $p_4^*(\phi_4, \theta_4)$. Each of those distributions has the same product form

$$p_i^*(\phi_i, \theta_i) \propto \exp(\kappa \cos(\theta_i - \mu_i)) \exp(\kappa \cos(2(\phi_i - m_i))), \tag{9}$$

where $\kappa = 6$. However, they differ in their mean values $\mu_i$ and $m_i$, see Table S7.

| i | $\mu_i$ | $m_i$ |
|---|---------|-------|
| 1 | $\frac{\pi}{2}$ | $\frac{\pi}{4}$ |
| 2 | $\frac{4\pi}{3}$ | $\frac{3\pi}{4}$ |
| 3 | $\frac{\pi}{2}$ | $\frac{3\pi}{4}$ |
| 4 | $\frac{\pi}{4}$ | $\frac{4\pi}{3}$ |

Table S7: Mean values for the mixture of von Mises distributions.

The dataset consists of transformed samples from this mixture distribution, where the transformation is given by

$$g(\theta, \phi) = (\sin(\theta)\cos(\phi), \sin(\theta)\sin(\phi), \cos(\theta))^T. \tag{10}$$

Thus, the Gram determinant of $g$ is $\det G_g = \sin(\theta)$.

**Training protocol and architectures:** We use the same protocol and architecture as for the thin spiral experiment.

**Metric:** To evaluate the performance, the Kolmogorov-Smirnov (KS) statistic was used. The KS statistic is defined as

$$KS = \sup_{x \in \mathcal{X}} |F(x) - G(x)|, \tag{11}$$

where $F$ and $G$ are the cumulative distribution functions associated with the probability densities $p(x)$ and $q(x)$ with domain $\mathcal{X}$, respectively. By definition, $KS \in [0, 1]$ and $KS = 0$ if and only if $p(x) = q(x)$ for almost every $x \in \mathcal{X}$.

**Results:** In Figure S2, we plot the target distributions in the first row (the true latent distribution for the sphere on the left, and samples for the circle on the right). In the second row, we show the learned latent distributions using the DNF. For that, we first divide the learned density $q_\sigma(x)$ with the normalization constant $q_\sigma(x|x)$ of the noise distribution used for the inflation (a standard Gaussian with $\sigma^2 = 0.01$) to obtain an estimate for $p(x)$, i.e. $p(x) = q(x)/q_\sigma(x|x)$. Second, we multiply this estimate with the corresponding Gram determinant of the true generating mapping.

Thus, for the case of a spiral and sphere, the DNF can be used to approximate the true Gram determinant almost exactly. This is also reflected in the small KS-values which we report in Table S8.

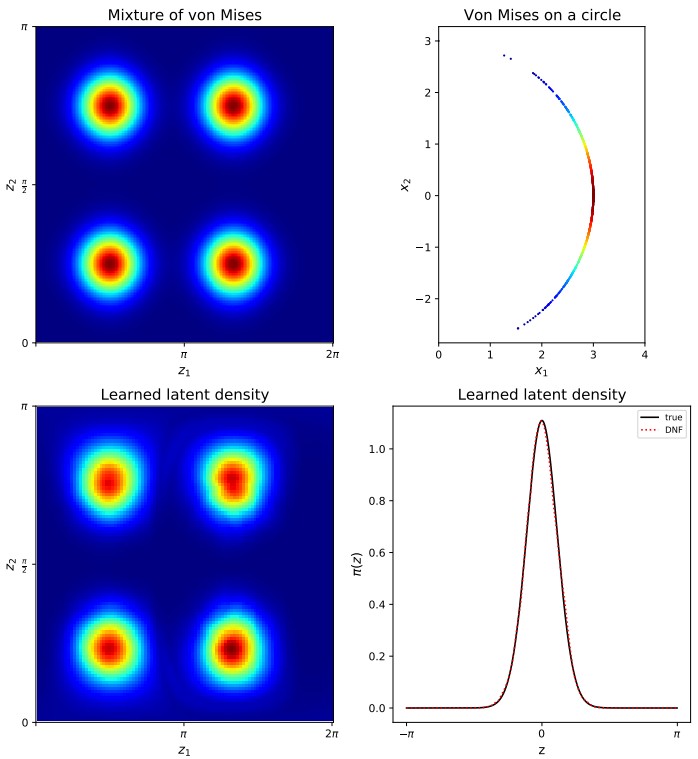

Figure S2: DNF with $\sigma^2 = 0.01$ on a sphere (left) and a circle (right).

|  | KS | |
| Model | circle | sphere |
| --- | --- | --- |
| inflation-deflation | **0.004** | **0.021** |
| DNF with $\sigma^2 = 0.01$ | 0.008 | 0.023 |

Table S8: KS values of DNF and inflation-deflation method on circle and sphere.

### C.3   DNF for probabilistic inference

**Dataset:** In [2], the graph of a function $f : \mathbb{R}^2 \to \mathbb{R}$,

$$f(z) = \exp(-0.1||z||) \sum_{ij} a_{ij} z_0^i z_1^j, \tag{12}$$

was considered. Further, points on the graph were rotated using a rotation matrix $R \in \mathbb{R}^3 \times \mathbb{R}^3$, i.e. the 2-dimensional manifold embedded in $\mathbb{R}^3$ consists of points $x = R(z_0, z_1, f(z))^T, z = (z_0, z_1)^T$. We refer to the appendix of [2] for the values of $R$ and $a_{ij}$. A density on that manifold was induced by a mixture of two Gaussians in the latent space conditioned on some parameter $\theta \in [0, 1]$,

$$p(z|\theta) = 0.6\mathcal{N}\left(z \middle| \begin{pmatrix} 1 \\ -1 \end{pmatrix}, 2^2 I\right)$$
$$+ 0.4\mathcal{N}\left(z \middle| \begin{pmatrix} -1 \\ 1 \end{pmatrix}, (0.6 + 0.4\theta)^2 I\right) \tag{13}$$

A training sample $x$ is generated by first sampling $\theta \sim \text{Uniform}(-1, 1)$, then generating latent variables $z \sim p(z|\theta)$, and finally setting $x = R(z_0, z_1, f(z))^T$. Like this, $10^5$ training examples were sampled.

**Training protocol:** As in [2], we train on 100 epochs with a batch size of 100 (i.e. in total $5 \cdot 10^4$ updating gradient steps). We use the AdamW ([12]) optimizer with annealing of the learning rate to 0 after 50 epochs using a cosine schedule ([11]).

**Architectures:** We use the same architecture as in [2], see Table S9. Note that $f_\psi$ does not depend on $\theta$ whereas the density learning NF $h_\phi$ is conditioned on the corresponding $\theta$ used to generate the training example.

| | | flow $f_\psi$ | | | flow $h_\phi$ | | |
|---|---|---|---|---|---|---|---|
| model | # levels | # couplings | coupling type | masking | # couplings | coupling type | masking |
| $\mathcal{M}$-flow/ DNF | 1 | 6 | spline with $B = 6, K = 10$ | lu | 4 | spline with $B = 6, K = 10$ | lu |

Table S9: Architecture for $\mathcal{M}-$flow and DNF on the polynomial surface dataset..

**Inference task:** Given a new sample $x$, the goal is to infer $\theta$ such that $\theta \sim p(\theta|x)$. From Bayes rules, we have that

$$p(\theta|x) \propto p(x|\theta)p(\theta) \tag{14}$$

such that given a model for $p(x|\theta)$ and a prior $p(\theta)$ (Uniform$(-1, 1)$ in this case), samples from $p(\theta|x)$ can be generated with a Markov Chain Monte Carlo (MCMC) sampler. In [2], an MCMC of length 5000, with a Gaussian proposal distribution with step size 0.15 and a burn in of 100 steps was used to generate posterior samples using 10 samples from the true density $p(x|\theta)$ for $\theta = 0$. Using the maximum mean discrepancy (MMD) [5], these samples were compared with MCMC samples using the true density $p(x|\theta)$.[1] In Table S10, we report the median MMD based on five runs with independent training data and initializations. The DNF clearly outperforms the $\mathcal{M}-$flow and performs similar to the $\mathcal{M}_e-$flow, a variant of the $\mathcal{M}-$flow where the encoder is not restricted to be invertible.

| Model | Posterior MMD |
|---|---|
| $\mathcal{M}-$flow | 0.017 |
| $\mathcal{M}_e-$flow | 0.007 |
| DNF | **0.005** |

Table S10: Median MMD (lower is better) out of 5 independent runs using different model-likelihoods ($\mathcal{M}-$flow ,$\mathcal{M}_e-$flow, DNF) for a MCMC sampler.

## C.4 CelebA-HQ

Different from the StyleGAN image manifolds, the dimensionality (if it exists) is not known for this dataset. In [2], $d$ was set to $512$ which resulted in a worse performance than a standard NF. One reason may be a bad choice for $d$. Another reason could be that the CelebA data-manifold is simply not a differentiable manifold or at least not describable by a single chart. Extending the DNF or $\mathcal{M}-$flow to allow for multiple charts is an interesting research question.

---

[1]This can be done because the Jacobian determinant of the manifold generating mapping does not depend on $\theta$, see the supplementary of [2] for more details.

The compare the DNF's performance with the $\mathcal{M}-$flow on such a real-world dataset, we trained an DNF using the same architecture and training settings as the $\mathcal{M}-$flow in [2][2].

In Figure S3, we show in the first 5 columns samples from the original dataset (top), $\mathcal{M}-$flow (second row), and DNF (last row). In the remaining 5 columns, we show how smoothly these models interpolate linearly in the latent space. For that, we linearly interpolate between two training images in latent space and display the corresponding trajectory in image space. We compare the FID and mean reconstruction error of the models in Table S11.

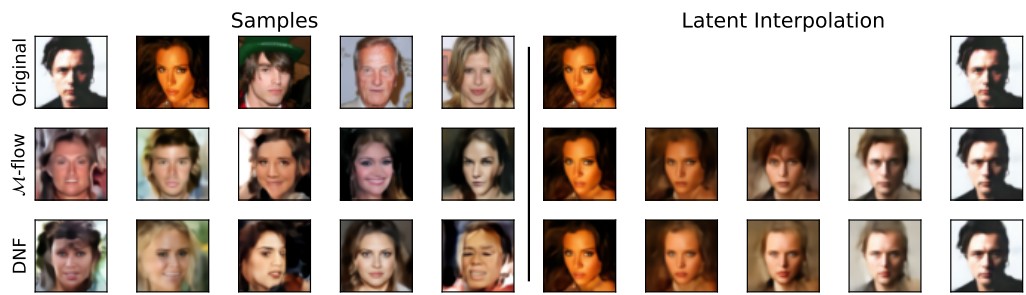

Figure S3: **Samples** from the CelebA dataset, $\mathcal{M}-$flow, and DNF (first 5 figures). **Latent interpolation** of $\mathcal{M}-$flow (middle) and DNF (bottom).

|  | CelebA | |
| --- | --- | --- |
| Model | FID | Mean reconstr. error |
| $\mathcal{M}-$flow | 38.07 | **831.5** |
| DNF | **34.14** | 858.1 |

Table S11: FID and mean reconstruction error of the $\mathcal{M}-$flow and DNF on the CelebA dataset. For both models, we set $d = 512$ and follow the original training protocol used in [2].