# OpenReview forum: "Denoising Normalizing Flow"
_NeurIPS.cc/2021/Conference — NeurIPS 2021 Poster_

### Official Review · Reviewer_xmqi · 2021-07-16

**Rating:** 6
**Confidence:** 4

**Summary:**

This paper proposes Denoising Normalizing Flows (DNFs), a method for accounting for low-dimensional manifold structure of data when using Normalizing Flows (NFs). The method builds on the recent work Horvat and Pfister ([18] in the manuscript), who showed that under some conditions, adding a small amount of Gaussian noise ("inflation") to data and then using a fully dimensional NF can recover the correct distribution on the manifold after a "deflation" step. The authors combine injective flows, mapping from low-to high dimensional space, with the inflation/deflation process, in an attempt to recover a low dimensional representation as well (as opposed to just using the inflation/deflation method); and properly accounting for the change-of-volume term during training (as opposed to the M-flow method of Brehmer and Cranmer, [9] in the manuscript).

**Ethical Concerns:**

None.

**Limitations And Societal Impact:**

Limitations and potential negative societal consequences are adequately discussed.

**Main Review:**

While I think the problem that this paper tackles is relevant and that the idea is nice, I found it to not be very well written and a bit hard to follow. I also have my doubts about the experimental results. I elaborate below.

1. I find the presentation for the objective hard to parse. Equation 13 is introduced as "the model", but does not really correspond to a density (proposition 1, which enables the density interpretation, is only presented later), and the fact that this equation avoids the injective change-of-volume term is not emphasized. The notation \tilde{u} is also confusing, as so far tilde had been used to denote added Gaussian noise. I had to read the section a couple times before I understood what the authors were trying to do. I do find this idea very interesting though, as an elegant way of avoiding having to compute the challenging log determinant by instead computing two easy-to-compute ones.

2. Part of the requirements for proposition 1 is that L_{DNF}(\theta) = 0, but this can actually never happen unless \sigma=0, since C(\psi) (equation 12) will in general not be 0 unless \sigma=0. The way I understand proposition 1 is as justifying using equation (13) rather than accounting for the injective change-of-volume-term; but if this condition can never be met, it throws the usefulness of proposition 1 into question. Since C(\psi) is arbitrarily chosen, would it not make sense to change it to E_{x \sim p(x)}[||x - r_\psi(x)||^2]?

3. I think a more detailed discussion of the benefits of DNFs as compared to the inflation/deflation method is warranted, as this baseline also manages to account for low-dimensional manifold support while remaining tractable. For a method so heavily inspired by this previous work, empirical comparisons against it should be carried out (I am aware that the inflation/deflation paper was mostly theoretical and presented only low-dimensional experiments; but still think the comparison should be made).

4. I find the results on the toy example in Figure 2 for all baselines to be suspiciously poor. I have personally trained standard NFs and M-flows on very similar 2-D examples with a 1-D manifold -- albeit not the same one -- and obtained much better (though not perfect) results. The authors do not specify which range of hyperparameters were considered for this example, and I suspect they might have not given a fair chance to the baselines by not tuning them enough.

5. Similarly, Table 1 shows DNFs achieve smaller reconstruction errors than M-flows. I also find this intriguing, as DNFs incorporate reconstruction error as a regularization term in the loss; thus achieving a tradeoff between this term and the KL term; whereas M-flows train the injective flow exclusively to minimize reconstruction error. Again, without a detailed explanation for why this might be happening, or at the very least verification that it wasn't a fluke, I can only be suspicious of the results.

6. I also think experiments on actual images rather than just naturalistic images would improve the paper.

Additional thoughts/questions/typos:

-Line 27: I don't see the relevance of the manifold being Riemannian: why is it having a metric relevant?

-Lines 36-38: the sentence is not properly formed.

-Equation 1: G_g should be defined, rather than its determinant.

-Lines 57-58: the statement " in generative modeling, it is assumed that the data x \in M is generated by a non-linear, injective transformation g...". This statement is too strong and should be toned down; this assumption is made only in injective flows, and as a simplifying assumption, as it implies -- unrealistically -- that the support is homeomorphic to R^d.

-Line 109: q_\sigma(\tilde{x} \cup N_x|x) is not defined, nor is it standard notation.

-Line 126 says q_\sigma(\tilde{x}|x) is a D-d dimensional Gaussian, and \tilde{x} is D-dimensional (in the algorithm box, D-dimensional noise is added).

-Line 171: The work of [1] is relevant when discussing overcoming topological constraints in flows.

-Line 213: neuronal -> neural

-Line 249: n-dimensional -> d-dimensional

-Line 261: tabular 5.2 -> Table 1

-Figure 2 and Table 1 have captions on top and on bottom, and it should be the other way around.

-Lines 307-308: You have not showed DNFs can be used for out-of-distribution detection nor for robustness, these claims should not be made without empirical backing, or should be clarified as hypotheticals, with at least some intuition as to why they might be true.

[1] Relaxing Bijectivity Constraints with Continuously Indexed Normalising Flows, Cornish et al., ICML 2020

======================================================================================================

EDIT 1 AFTER REBUTTAL

I have read the other reviews as well as the author's rebuttal, and thank the authors for their additional clarifications. While some of my doubts about some experiments have been resolved, and I agree that my initial suggestion for $C(\psi)$ was not good, I still have some of my original concerns: The assumption for proposition 1 to hold is that noise is added in the normal direction to $x$ rather than Gaussian noise as the authors add in practice; which effectively means there is no guarantee that the determinant can actually be approximated. Additionally, like I mentioned in my response to the author's rebuttal, adding noise just to $v$ seems like a good way of ensuring that $x$ can indeed be recovered from $\tilde{x}$ -- which is not ensured by the Gaussian noise they use -- and thus a more natural way of adding noise and taking advantage of the flow structure than what the authors currently do. This modification to the author's approach seems to me like a more natural way to proceed, and would need to either be shown the modification working, or an argument for why it doesn't before increasing my score, which I am thus leaving unchanged for now.

======================================================================================================

EDIT 2 AFTER REBUTTAL

I have read the additional verification performed by the authors on reconstruction errors and have decided to increase my score. While I am still concerned about the theoretical claims not applying to the proposed choice of inflation noise and that exploring other alternatives would further strengthen the paper, I am no longer skeptical of the empirical results, which I actually find interesting and intriguing. I would still strongly encourage the authors to both:

1. More clearly specify that the Gaussian noise they use for inflation does not satisfy the requirements for their theorem, and that the theorem should thus be interpreted as giving intuition, rather than providing guarantees for the proposed empirical method.

2. Either try to explain why $\mathcal{M}$-flows achieve worse reconstruction error than the proposed method, or highlight this as intriguing behaviour. I still believe these results are a bit puzzling as $\mathcal{M}$-flows directly minimize reconstruction while the proposed method only includes a similar term as a component of its loss function.

======================================================================================================

**Time Spent Reviewing:**

5

---

### Official Review · Reviewer_8Mvk · 2021-07-16

**Rating:** 6
**Confidence:** 4

**Summary:**

This paper is motivated by the desire to circumvent a troublesome assumption in Normalizing Flows (NFs) on the function $f$ that produces the pushforward measure --- that $f$ must be strictly invertible. This is difficult for generative models, since the dimension of the latent space $d$ is usually much smaller than that of the target data space $D$, i.e. d << D.

To address this, the authors train a transformation $f:\mathbb{R}^D \rightarrow \mathbb{R}^D$ as usually done in a NF approach, but separate the latent variable $z = f^{-1}(x)$ into two orthogonal components $u, v$, consisting of the first $d$ and remaining $D-d$ coordinates of $z$. Intuitively, $u$ should contain the "manifold" information of $x$, and $v$ should capture any "off-manifold" noise. This is encouraged by adding a reconstruction loss to the $u$ component of the latent space. To allow for greater complexity of the latent variable $u$, a second NF transformation $h$ is applied to u, i.e. $u = h(\tilde{u})$. In addition, the authors add noise robustness by adding Gaussian noise to $x$, i.e. $\tilde(x) = x + \epsilon$, and reconstruct $x$ --- much like with Denoising Autoencoders, hence the name.


**Limitations And Societal Impact:**

As a highly theoretical work, there are no clear societal impacts of this paper. I am, however, concerned with the applicability of this work in datasets where the manifold assumption does not hold. As mentioned, I think that an exploration of the performance of DNFs in more realistic datasets will help explore this question.

**Main Review:**

There are two main aspects to the denoising NF method:
1. The latent variable $z$ is decomposed into two parts $(u, v)$, where one part ($u$) is encouraged to encode the signal, and the other part ($v$) is encouraged to encode the noise.
2. Noise is added to the original training sample $x$, and the proposed method is trained to denoise the noisy $\tilde{x}$, like in Denoising Autoencoders.

The authors disclose that part 1) has been attempted in part by a prior paper, but the addition of a "reconstruction loss" as motivated by part 2) is crucial, in my opinion, and makes the contribution in this paper unique.

Moreover, the method is well-motivated --- there is currently no good way to apply NFs to generative models, even though NFs are a promising candidate for deep generative modeling. Moreover, the technique is straightforward and technically sound.

However, I do have three issues with the paper as it stands:

1. I find the experiments to be somewhat lacking. The manifold assumption, while powerful, does not usually hold exactly in real-world data. Both experiments in the main text (synthetic spiral, StyleGAN image manifold) involve data that lie strictly on a manifold. The supplemental material includes an experiment with the CelebA faces dataset. However, the proposed method is only compared to one other method (M-flow [1]). I would like to see how DNFs perform against GANs and VAEs on more realistic datasets.
2. The connection to Denoising Autoencoders is nice, but it seems that Experiment C.1 (in the Supplementary Material) suggests that the noise parameter $\sigma^2$ is not that crucial to the performance of the model. It would be nice to see further discussion of the importance of the Denoising aspect of the method.
3. I found the paper well written overall, but a crucial detail seems to be incorrect. In Eq. 14, it appears that the loss $\mathcal{L}_{DNF}(\theta)$ is the (weighted) sum of the reconstruction term $C(\psi)$ and a KL divergence term between $q_\sigma$ and $q_\theta$. However, in the algorithm, the KL divergence term is replaced by a negative log probability term $-\log q_\theta$. Is Eq. 14 simply a typo, or am I misunderstanding something? (Furthermore, the $p_v(v_i)$ term in the negative log probability seems to be missing a $\log$.) To my knowledge, it is not possible to evaluate the KL divergence between $q_\sigma$ and $q_\theta$, as $q_\sigma(\tilde(x)) = q_\sigma(\tilde(x) | x) p(x)$, and $p(x)$ is not known.

**Time Spent Reviewing:**

10

---

### Official Review · Reviewer_TzDc · 2021-07-19

**Rating:** 7
**Confidence:** 3

**Summary:**

Authors propose a novel normalizing-flow-based method for simultaneous manifold and probability-density-on-the-manifold learning. Similarly to previous methods addressing this problem, the dimensionality of the manifold learned needs to be set by the user before training and is not learned as part of the process, albeit there exist other methods for estimating it. The advantage of the proposed method is the single-step training procedure, in contrast to the multi-step training procedure of the other competitive method in this domain. I consider it a good contribution towards building a normalizing flow based representation learning toolkit.

**Limitations And Societal Impact:**

Authors did a fair job discussing the limitations of their method.

Societal impact: N/A


**Main Review:**

Good paper: the core idea seems to be well developed and explained, decent related work section, good mathematical notation. My main criticism is that I would encourage the authors to expand the empirical evaluation section.

*Originality:*
- To the best of my knowledge the method is novel, although based to some extent on an existing insight.
- While it might be true that “the inflation approach” for learning manifold density with normalizing flows was discussed in particular depth in [18], authors should acknowledge that the idea of applying it in exactly this context was present in the literature earlier than [18] --- one earlier work to point out is [22], but it’s possible there exist works earlier than this one. I think it’s worth acknowledging it properly for the sake of proper credit assignment in the community.
- Apart from this detail, I find the related work section quite well developed and Fig 1 does a good job explaining how the work differs from some of the other most related methods.
- One other interesting work to cite in this context, especially regarding the detection of the manifold dimensionality, is arxiv:2006.08777.

*Quality:*
- I believe the work is technically sound, the core idea seems to be well developed and explained, and most of the conclusions regarding the empirical evaluation of the proposed method are substantiated.
- I haven’t caught any apparent errors but due to the heavy reviewing workload I could not commit enough time to carefully check the mathematical arguments, especially the ones in [18], which seems to not have been published in any proceedings before and hence I assume did not undergo peer review.
- [22] learn to utilize NFs to sample from the manifold, but it’s not useful for representation learning, due to the lack of ability to learn latent representations of dimensionality lower than the data space dimensionality. Both [9] and the proposed method overcome those difficulties and showcase their utility for representation learning. [9] also showcases its utility for probabilistic inference and I’m convinced this work could also evaluate itself in such a setting which would enrich the paper, and currently in my opinion somewhat modest empirical evaluation section (modest compared to [9] which is quite rich so that might not be an entirely fair point of comparison). However, given that the key metric the authors are comparing on is sample quality, it might be worthwhile to compare the proposed method also to Softflow --- keeping the architecture fixed as much as possible across models, it would provide an interesting comparison of what is the sample-quality price to be paid for the representation learning properties of the proposed method.
- One of the key goals of the proposed method is to learn the on-manifold density, but the performance in that aspect of the task is barely evaluated empirically --- the only part of the evaluation that touches upon that aspect is the limited visual comparison offered in Figure 2. In all honesty, it is not immediately clear to me what would make for a good protocol to empirically evaluate this aspect of the method, so I am sympathetic to the fact it’s not studied better in the current version of the paper. However, I would encourage authors to think about this.
One vague idea that comes to my mind is: for some synthetic toy problem, make use of the knowledge of the underlying manifold to define a hand-crafted diffeomorphic transform between the learned manifold and the true manifold (since the learned one obviously will not be perfect), and then compute e.g. KL divergence between those two d-dimensional probability densities. Is this a viable direction?
- L117-118: I’m wondering about the effect of using the full Gaussian noise on the on-manifold density in cases when the on-manifold density changes quickly, e.g. the on-manifold density oscillates with a high frequency. Could authors comment on this please?


*Clarity:*
- Yes, the submission is mostly clearly & well written. There were very few points when I had to reread or pause due to the quality of writing.
- Formatting wise, I find the “Research question” and “Question 1” right next to each other a bit too much, but overall I appreciate the use of paragraph captions.
- Is “the characteristic function” \mathcal{X} in L235-236 intended to mean an indicator function? This isn't immediately clear to me.
- Fig. 2 goes outside of allowed margins.
- Formatting of the citation: ([1]) -> [1], ([1], [2]) -> [1, 2]

*Significance:*
- I consider this paper a good contribution towards building a normalizing flow based representation learning toolkit, and I think it might find downstream applications in the future, especially given its simplified training procedure.


**Time Spent Reviewing:**

6

---

### Author Response · Authors · 2021-08-09
**Answer to reviews**

We sincerely thank all reviewers **(R1,R2,R3)** for reading our manuscript carefully and for giving valuable feedback. We are convinced that incorporating them in the final version will significantly improve this work. In the following, we answer some of the most pressing questions and comments:

* **Empirical evaluation (R1,R2,R3):** All reviewers agree that the empirical evaluation section could be expanded **(R1)** to include more realistic datasets **(R2,R3)**, and better evaluate the learned on-manifold densitiy **(R1)**, e.g. by comparing to the inflation/deflation method on which the Denoising Normalizing Flow (DNF) heavily depends on **(R3)**. Also, utilizing the DNF for probabilistic inference (as in reference [9] in the paper) would enrich the work **(R1)**.
_Due to limited computational resources, we suggest for the camera-ready version the following:_

  1. We train the DNF on the von Mises on a circle experiment conducted in the inflation/deflation paper (reference [18] in the paper) which includes calculating the Kolmogorov-Smirnov statistics in order to evaluate the on-manifold density and allow a direct comparison with [18].
  2. We utilize the DNF to perform probabilistic inference following the protocol used in the $\mathcal{M}-$flow paper ([9]).



* **Benchmarks (R1,R2):** We agree that a more in-depth comparison with related models (such as SoftFlow and GANs) would be interesting. We will leave such a comparison for future work.


* **Is “the characteristic function” $\mathcal{X}$ in L235-236 intended to mean an indicator function? (R1) :** 	Indeed, it was intended to mean an indicator function. The idea is to only consider points that have a small $v$ value to be in-manifold as the off-manifold direction is encoded in $v$. _We will make this clearer for the final version._


* **Effect of using the full Gaussian noise on the on-manifold density in the case when the on-manifold density changes quickly (R1):** The reviewer is correct in pointing out that if the on-manifold density changes too quickly, then it can not be learned. Loosely speaking, for the on-manifold density to be learnable, we need the second derivative of the on-manifold density to be much smaller than $1/\sigma^2$ where  $\sigma$ denotes the standard deviation of the additive noise in the inflation phase. A rigorous condition can be found in Proposition 6 in [18].


* **The discrepancy between $\mathcal{L}_{\text{DNF}}(\theta)$ and the algorithm (R2):** We thank the reviewer for this comment. There was indeed a typo in the display of the algorithm. $p_v(v_i)$ should indeed be replaced by $\log(p_v(v_i))$. This typo was not present in the code, so the numerical experiments are correct.
 We also agree that it is indeed confusing to express $L_{DNF}$ in Eq. 14 as a function of the $D_{KL}$ between $q_\sigma$ and $q_\theta$ (this is needed for proposition 1) while ignoring the $q_\sigma$ entropy term in the algorithmic box. To remove this confusion, we will denote the cost function in the algorithmic box as $L_{DNF}^{'}$ and stress that it only differs from  $L_{DNF}$ by a constant term (i.e the  $q_\sigma$ entropy term which is independent of $\theta$). Note that this distinction does not affect numerical simulations.


* **Discussion on the denoising aspect (R2,R3):** The additional experiment in the supplementary using $\sigma^2=0$ indeed raises the question on the necessity of $\sigma^2>0$. We will repeat this experiment with the thin spiral where the effect of noise will arguably have a greater impact (as the Jacobian determinant of the outer flow $f$ eventually degenerates). One reason why the effect of adding noise on the StyleGan 2d image manifold is less significant might be the dequantization in the pre-processing which is, essentially, an inflation of the original data-manifold with a small amount of noise (i.e. the resulting inflated manifold is not exactly a 2-dimensional manifold anymore). Also, $\sigma$ is an additional hyperparameter we haven't tuned so far and might increase the effect of the denoising.

  _To further strengthen the evidence of the effect of the denoising, we will increase the number of seeds used for the StyleGan 2d image manifold experiments, and add the DNF with $\sigma^2=0$ on the thin spiral in the supplementary._


* **Suspiciously poor results on all baselines for the thin spiral toy example (R3):** For the $\mathcal{M}-$flow, we have used the same settings as for the circle experiment in [9]. There, noise was added to the circle which may explain the difference in performance. We also tried different settings (inspired by the $\mathcal{M}-$flow experiment on a 2-dimensional manifold embedded in $\mathbb{R}^3$) with no significant difference no matter the training scheme (sequential or alternating).

  For the Normalizing Flow (NF), everything else but a degenerative result would be very surprising as the Jacobian determinant eventually degenerates.

  For the Probabilistic Auto Encoder (PAE) and Variational Auto Encoder (VAE), we will double-check the implemented density evaluation method.


* **$L_{DNF}(\theta)=0$ never happens (R3):** We disagree with the statement that $L_{\text{DNF}}(\theta) = 0$ can never happen unless $\sigma = 0$. Indeed, if the inflation process ($\tilde{x} = x + \sigma\epsilon$) is such that $\tilde{x} \in N_x$ where $N_x$ is the normal space at $x$ (see the $Q-$normal separability condition in definition 4 and theorem 5 of [18]) then there exists a denoising function $r_\psi(\tilde{x})$ such that $x = r_\psi(\tilde{x})$. Therefore the regularisation term $C(\psi)$ can be zero even if $\sigma$ is not zero.

  Regarding the definition of $C(\psi)$, we don’t believe it is a good idea to change it to $E_{x\sim p(x)}[||x-r_\psi(x) ||^2]$ as suggested by the reviewer. This new definition of $C(\psi)$ would make it independent of the inflation whereas the denoising function $r_\psi$ has to precisely learn to undo the noise inflation.


* **Suspicion on the fact that DNF achieves smaller reconstruction error than $\mathcal{M}-$flow (R3):** One of the benefits of having a single loss function is that the corresponding weights do change the gradients and not only scale them. The DNF's hyperparameter $\lambda$ for the reconstruction error interpolates between density estimation ($\lambda=0$) and manifold learning ($\lambda \to \infty$). In contrast, a pre-factor of the $\mathcal{M}-$flow simply scales the gradient. Thus, although the reconstruction term $\mathcal{C}(\psi)$ in the DNF loss is indeed a regularizer, choosing $\lambda=1000$ (as it was set for the StyleGan image-manifolds) is a great incentive to keep the reconstruction error small (arguably a greater incentive than training on just the reconstruction error).

  _To further reduce the chance of fluke, we will increase the number of seeds used for the StyleGan 2d image manifold experiments. If time allows, we will do the same for the $64-$dimensional case._

---

> ### Comment · Reviewer_xmqi · 2021-08-11
> **Response to the rebuttal**
>
> I thank the authors for their rebuttal.
>
> 1. On $L_{DNF}(\theta)=0$ not happening: First, you are correct that the modification I proposed to $C(\psi)$ in my review indeed misses the point, however, I do not actually agree that $L_{DNF}(\theta)=0$ can happen. You start by saying "if the inflation process is such that $\tilde{x} \in N_x$", which happens with probability 0. The definition and theorem you allude to in [18] have this assumption, which requires that the added noise for inflation be supported only in the normal direction to the manifold at $x$, rather than $\mathbb{R}^D$ as in the fully-dimensional case as used in the paper. For example, in Figure 2 (left) of [18], one can see this: one can recover a point on the black circle from one on the red area assuming the second one was generated from the former only if the noise is assumed to be orthogonal to the tangent line at the circle. If 2-dimensional Gaussian noise is used instead, the probability of lying on this affine space will be 0. I am aware that the authors of [18] argue that in high dimensions, having fully dimensional noise approximates noise in the normal space; but this is equivalent to saying $L_{DNF}(\theta)$ can be close to 0, not that it can be 0. I wonder though, if in the spirit of my originally proposed modification but in a way that actually accounts for inflation, one could just modify the definition of $q_\sigma(\tilde{x}|x)$ to be defined as only adding noise on $v$? That is, what happens if instead of obtaining $\tilde{x}$ as $\tilde{x} = x + \sigma \epsilon$ for $\epsilon \sim \mathcal{N}(0, I_D)$, one were to obtain $\tilde{x}$ by $(u, v) = f^{-1}_{\psi}(x)$, $\tilde{v} = v + \sigma \epsilon$ now with $\epsilon$ being $(D-d)$-dimensional, and then define $\tilde{x} = f_\psi(u, \tilde{v})$, all while keeping $C(\psi)$ as originally defined? In this setting I can at least see how the noise can actually be ignored, although I am open to another potential issue being highlighted with this proposal.
>
> 2. On suspiciously poor spiral toy example results: Thank you for the clarification, I believe the highlighted differences likely explain the results.
>
> 3. On reconstruction error smaller than $\mathcal{M}$-flows: I am not sure I agree that one should expect the KL term to act as a regularizer, thus resulting in smaller reconstruction error than directly optimizing the reconstruction error. I would intuitively just expect similar reconstruction errors for large values of $\lambda$

---

> > ### Author Response · Authors · 2021-08-13
> > **Response to response**
> >
> > We thank the reviewer for the quick and precise response.
> >
> >  1. We belive that there is a misunderstanding on the reviewers side regarding the statement of Proposition 1. There, one condition is that equation (8) holds, i.e.  $q_{\sigma}(\tilde{x})=p(x)q_{\sigma}(\tilde{x}|x)$. This condition implicitely assumes that the noise is only added in the manifold's normal space (in such a way that the $Q-$normal reachability condition is fulfilled). Therefore, $L_{DNF}$ can be equal to zero, but only when the added noise is in the normal space. When the noise is not normal (e.g. full Gaussian noise), then   $L_{DNF}\neq 0$. _We will make this normal space assumption explicit in Proposition 1. for the final version._
> >
> > 	However, we thank the reviewer for the nice idea to add the noise only in the $v$-component. We also believe that a (pre-trained) DNF can be used to sample from the manifold's normal space and improve the DNF's performance. We believe that this is an interesting research direction to pursue.
> >
> >  2.  For sake of transparency, we wanted to inform the reviewer that we double-checked the implementation of the PAE. In the current version of the paper, we estimated the density for the PAE according to the implementation in [9], and we found a mistake there. We now implemented the PAE's original density estimator (reference [8] in the paper). This improved the PAE's performance significantly. However, the DNF still outperforms the PAE in terms of learning the on-manifold density. Thus, this does not impact the message of the paragraph.
> >
> >  3. We now fully understand the reviewers doubt. Indeed, it is not intuitive why the $D_{KL}$-term should act as a regularizer. We admit that the $\mathcal{M}-$flow might yield similar results in terms of reconstruction error with a thorough hyperparameter search. Nevertheless, there might be a principled reason for why the $D_{KL}-$term helps to regularize $f$. We thank the reviewer for sparking this very interesting research question.
> >
> >     Note that we have used the original implementation of the M-flow on the StyleGan 2d dataset (for which a hyperparameter search was already conducted). For the DNF, we used the very same hyperparameters except changing the pre-factor of the reconstruction error from 1 to 1000 (doing the same for the M-flow leads to very poor results indicating its hyperparameter sensitivity).  We, therefore, think that this original M-flow still serves as the right benchmark for the DNF, even without a further hyperparameter search.

---

> ### Author Response · Authors · 2021-08-16
> **Discussion on the denoising aspect (R2,R3):**
>
> We wanted to inform the reviewers that we trained the DNF with $\sigma^2=0$ on the thin spiral, and obtained, as expected, a degenerated result (similar to the NF's result). This emphasizes the importance of the denoising aspect.

---

> ### Author Response · Authors · 2021-08-24
> **Chance of fluke (R3) and discussion on the denoising aspect (R2,R3)**
>
>  As announced, we have increased the number of seeds used to train the DNF (with $\sigma^2=0$ and $\sigma^2=0.1$) and the $\mathcal{M}-$flow on the StyleGan 2d dataset. Please find below the results using 10 seeds and removing the corresponding best and worst-case results (i.e. using 8 seeds in total for the mean and standard deviation). We hope that this helps to convince **R3** that the DNF's superior performance on the reconstruction error was not a fluke.
>
> Model     $\qquad \qquad \quad $                    FID                      $\qquad \qquad \qquad $                  Rec. Error
>
> DNF $\sigma^2 =0$        $\qquad \quad$       $4.38\pm 0.26$      $\qquad \quad$                           $225.78 \pm 14.4$
>
> DNF $\sigma^2 =0.1$        $\qquad $       $4.42\pm 0.2$      $\qquad \quad$                           $224.2 \pm 20.6$
>
> $\mathcal{M}-$flow       $\qquad \qquad$       $4.85\pm 0.14$      $\qquad \quad$                           $309.32 \pm 6$
>
> Given these results, we will update the discussion on the denoising aspect (Section C.1 in the supplementary) and emphasize that the inflation is crucial for data living exactly on a low-dimensional manifold.

---

> ### Author Response · Authors · 2021-08-31
> **Learning the on-manifold density (R1,R3):**
>
> We trained the DNF on the von Mises distribution on the circle used in [18] to evaluate the quality of the learned on-manifold density. In [18], the Kolmogorov-Smirnov (KS) statistic was used to measure the discrepancy of the learned- and original density. The KS-value is always between 0 and 1. However, 0 is only achieved if and only if the two densities are identical. With our method (using the same settings as for the thin-spiral experiment), we get a KS-value of $\approx 0.008$ which is only marginally worse than the method proposed in [18] (KS-value of $\approx0.004$). This shows that indeed the on-manifold density was well learned. On top of that, the DNF learned the manifold (i.e. the latent $v$-component for points in the manifold is very close to $0$ whereas for points off the manifold the $v$-component is significantly higher). Remarkably, we trained the DNF using Gaussian noise, i.e. we have not used any information of the manifold but its dimensionality. Thus, equation (15) of our paper can, under certain conditions, serve as a great approximation even when using Gaussian noise.
>
> Therefore, in total, the DNF extends the work of [18] by not only learning the on-manifold density on the circle but also learning the manifold.

---

> > ### Comment · Reviewer_xmqi · 2021-08-31
> > **On KS test**
> >
> > I thank the authors for the additional experiment. Unless something not standard is being done however, it is not true that one should compare KS values directly. The KS test statistic measures a discrepancy between the true data-generating distribution and the empirical CDF obtained from a sample, and is thus intrinsically random. The standard thing to do is to report p-values rather than directly comparing the statistics, and from the numbers shared above, I would suspect both tests (i.e. for [18] and the proposed method) cannot reject the null hypothesis with statistical significance. Could you please either provide p-values, or explain why these are not being provided?

---

> > > ### Author Response · Authors · 2021-08-31
> > > **Answer to comment**
> > >
> > > We thank the reviewer for this comment. Indeed, we are using the KS statistics in a non-standard way and we realize that we didn’t make it clear enough.
> > >
> > > We are using the KS statistic as a distance measure between two parametric distributions and not between an empirical distribution and the reference distribution. Indeed,  both the DNF and the method proposed in [18] have full access to the estimated probability density function which is given parametrically. As a consequence, the KS statistic is not random (and therefore it doesn’t really make sense to use p-values here).  So, in order to compute the KS statistics, 100 equidistant points in the domain (in this case $[-\pi,\pi]$) were selected in [18] starting from $-\pi$ and ending with $\pi$.
> > >
> > > Our goal was to show that the estimated probability distribution from the DNF is actually very close to the ground truth distribution which is confirmed by the fact that the KS = 0.008. This also matches our observation when plotting the induced latent distribution against the true Von Mises distribution: they almost match perfectly.
> > >
> > > As a final remark, in [18] for each gradient step a new set of points were sampled whereas we trained the DNF on a fixed number of $10000$ samples. Arguably, the DNF's performance can be further increased when using the same training protocol.

---

> > > > ### Comment · Reviewer_xmqi · 2021-08-31
> > > > **KS discussion**
> > > >
> > > > I see, this makes sense, thank you for the clarification. While I agree that this result points towards the proposed method outperforming [18], I still think more challenging examples than the von Mises distribution are required to establish this empirically.

---

> > > > > ### Author Response · Authors · 2021-09-01
> > > > > **A more challenging example**
> > > > >
> > > > > We repeated the experiment on the mixture of von Mises distributions on a sphere (see example 5.2 [18] for more details). Again, we have found that the DNF learned the density similar to [18] which is reflected by $KS\approx 0.023$ (while [18] has a KS-value of $\approx 0.021$). Also, the $v$-component encoded the manifold (e.g. points that have a distance of $0.01$ to the manifold have, on average, a 15-times larger $v-$component than points directly on the manifold.)
> > > > >
> > > > > Please note that we don't claim to outperform [18]. The method proposed in [18] only learns the density on the manifold and not the manifold itself. Therefore, we see the DNF as an extension of [18] overcoming this limitation.

---

> ### Author Response · Authors · 2021-09-03
> **DNF for probabilistic inference (R1):**
>
> We wanted to confirm the intuition of R1 that the DNF can be used for probabilistic inference. We have trained the DNF on the polynomial surface dataset (see [9] for more details) which depends on a parameter $\theta$. To evaluate the inference performance, the maximum mean discrepancy (MMD) between posterior samples from the true- and model likelihood was calculated (the lower the better). The corresponding samples were obtained by an MCMC sampler based on a Gaussian kernel (see [9] for more details). In [9], the best model (a variant of the $M-$flow where the encoder is not restricted to be invertible) achieves a median MMD of $0.007$ out of 5 runs (see table 2). The vanilla $M-$flow achieves a median MMD of $0.020$. With our method, we got a median MMD of $0.005$. This makes us confident that the DNF can indeed be used for downstream applications (especially given its simplified training procedure, as mentioned by R1).

---

### Decision · Program_Chairs · 2021-09-27

**Decision:**

Accept (Poster)

**Comment:**

The paper proposes an NF with simultaneous manifold and density-on-manifold learning.  Inflation noise is added to make p(x) smooth and the reconstruction regularizer makes the first d variables independent from noise and hence extracts information of clean data.

The proposed method is well-motivated and shows good performance.  We expect that the authors revise the paper according to the discussion made with reviewers.